# YOLOv10: Real-Time End-to-End Object Detection

**Ao Wang**[1]    **Hui Chen**[2*]   **Lihao Liu**[1]    **Kai Chen**[1]
**Zijia Lin**[1]    **Jungong Han**[3]    **Guiguang Ding**[1*]
[1]School of Software, Tsinghua University    [2]BNRist, Tsinghua University
[3]Department of Automation, Tsinghua University
wa22@mails.tsinghua.edu.cn huichen@mail.tsinghua.edu.cn linzijia07@tsinghua.org.cn
{louisliu2048,chenkai2010.9,jungonghan77}@gmail.com dinggg@tsinghua.edu.cn

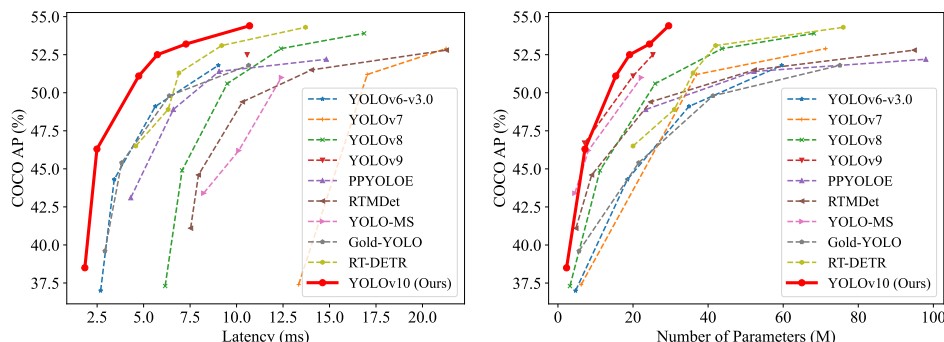

Figure 1: Comparisons with others in terms of latency-accuracy (left) and size-accuracy (right) trade-offs. We measure the end-to-end latency using the official pre-trained models.

## Abstract

Over the past years, YOLOs have emerged as the predominant paradigm in the field of real-time object detection owing to their effective balance between computational cost and detection performance. Researchers have explored the architectural designs, optimization objectives, data augmentation strategies, and others for YOLOs, achieving notable progress. However, the reliance on the non-maximum suppression (NMS) for post-processing hampers the end-to-end deployment of YOLOs and adversely impacts the inference latency. Besides, the design of various components in YOLOs lacks the comprehensive and thorough inspection, resulting in noticeable computational redundancy and limiting the model's capability. It renders the suboptimal efficiency, along with considerable potential for performance improvements. In this work, we aim to further advance the performance-efficiency boundary of YOLOs from both the post-processing and the model architecture. To this end, we first present the consistent dual assignments for NMS-free training of YOLOs, which brings the competitive performance and low inference latency simultaneously. Moreover, we introduce the holistic efficiency-accuracy driven model design strategy for YOLOs. We comprehensively optimize various components of YOLOs from both the efficiency and accuracy perspectives, which greatly reduces the computational overhead and enhances the capability. The outcome of our effort is a new generation of YOLO series for real-time end-to-end object detection, dubbed YOLOv10. Extensive experiments show that YOLOv10 achieves the state-of-the-art performance and efficiency across various model scales. For

---

*Corresponding author.

38th Conference on Neural Information Processing Systems (NeurIPS 2024).

example, our YOLOv10-S is 1.8× faster than RT-DETR-R18 under the similar AP on COCO, meanwhile enjoying 2.8× smaller number of parameters and FLOPs. Compared with YOLOv9-C, YOLOv10-B has 46% less latency and 25% fewer parameters for the same performance. Code and models are available at https://github.com/THU-MIG/yolov10.

# 1 Introduction

Real-time object detection has always been a focal point of research in the area of computer vision, which aims to accurately predict the categories and positions of objects in an image under low latency. It is widely adopted in various practical applications, including autonomous driving [3], robot navigation [12], and object tracking [72], *etc*. In recent years, researchers have concentrated on devising CNN-based object detectors to achieve real-time detection [19, 23, 48, 49, 50, 57, 13]. Among them, YOLOs have gained increasing popularity due to their adept balance between performance and efficiency [2, 20, 29, 20, 21, 65, 60, 70, 8, 71, 17, 29]. The detection pipeline of YOLOs consists of two parts: the model forward process and the NMS post-processing. However, both of them still have deficiencies, resulting in suboptimal accuracy-latency boundaries.

Specifically, YOLOs usually employ one-to-many label assignment strategy during training, whereby one ground-truth object corresponds to multiple positive samples. Despite yielding superior performance, this approach necessitates NMS to select the best positive prediction during inference. This slows down the inference speed and renders the performance sensitive to the hyperparameters of NMS, thereby preventing YOLOs from achieving optimal end-to-end deployment [78]. One line to tackle this issue is to adopt the recently introduced end-to-end DETR architectures [4, 81, 73, 30, 36, 42, 67]. For example, RT-DETR [78] presents an efficient hybrid encoder and uncertainty-minimal query selection, propelling DETRs into the realm of real-time applications. Nevertheless, when considering only the forward process of model during deployment, the efficiency of the DETRs still has room for improvements compared with YOLOs. Another line is to explore end-to-end detection for CNN-based detectors, which typically leverages one-to-one assignment strategies to suppress the redundant predictions [6, 55, 66, 80, 17]. However, they usually introduce additional inference overhead or achieve suboptimal performance for YOLOs.

Furthermore, the model architecture design remains a fundamental challenge for YOLOs, which exhibits an important impact on the accuracy and speed [50, 17, 71, 8]. To achieve more efficient and effective model architectures, researchers have explored different design strategies. Various primary computational units are presented for the backbone to enhance the feature extraction ability, including DarkNet [48, 49, 50], CSPNet [2], EfficientRep [29] and ELAN [62, 64], *etc*. For the neck, PAN [37], BiC [29], GD [60] and RepGFPN [71], *etc*., are explored to enhance the multi-scale feature fusion. Besides, model scaling strategies [62, 61] and re-parameterization [11, 29] techniques are also investigated. While these efforts have achieved notable advancements, a comprehensive inspection for various components in YOLOs from both the efficiency and accuracy perspectives is still lacking. As a result, there still exists considerable computational redundancy within YOLOs, leading to inefficient parameter utilization and suboptimal efficiency. Besides, the resulting constrained model capability also leads to inferior performance, leaving ample room for accuracy improvements.

In this work, we aim to address these issues and further advance the accuracy-speed boundaries of YOLOs. We target both the post-processing and the model architecture throughout the detection pipeline. To this end, we first tackle the problem of redundant predictions in the post-processing by presenting a consistent dual assignments strategy for NMS-free YOLOs with the dual label assignments and consistent matching metric. It allows the model to enjoy rich and harmonious supervision during training while eliminating the need for NMS during inference, leading to competitive performance with high efficiency. Secondly, we propose the holistic efficiency-accuracy driven model design strategy for the model architecture by performing the comprehensive inspection for various components in YOLOs. For efficiency, we propose the lightweight classification head, spatial-channel decoupled downsampling, and rank-guided block design, to reduce the manifested computational redundancy and achieve more efficient architecture. For accuracy, we explore the large-kernel convolution and present the effective partial self-attention module to enhance the model capability, harnessing the potential for performance improvements under low cost.

Based on these approaches, we succeed in achieving a new family of real-time end-to-end detectors with different model scales, *i.e.*, YOLOv10-N / S / M / B / L / X. Extensive experiments on standard benchmarks for object detection, *i.e.*, COCO [35], demonstrate that our YOLOv10 can significantly outperform previous state-of-the-art models in terms of computation-accuracy trade-offs across various model scales. As shown in Fig. 1, our YOLOv10-S / X are $1.8\times$ / $1.3\times$ faster than RT-DETR-R18 / R101, respectively, under the similar performance. Compared with YOLOv9-C, YOLOv10-B achieves a 46% reduction in latency with the same performance. Moreover, YOLOv10 exhibits highly efficient parameter utilization. Our YOLOv10-L / X outperforms YOLOv8-L / X by 0.3 AP and 0.5 AP, with $1.8\times$ and $2.3\times$ smaller number of parameters, respectively. YOLOv10-M achieves the similar AP compared with YOLOv9-M / YOLO-MS, with 23% / 31% fewer parameters, respectively. We hope that our work can inspire further studies and advancements in the field.

## 2 Related Work

**Real-time object detectors.** Real-time object detection aims to classify and locate objects under low latency, which is crucial for real-world applications. Over the past years, substantial efforts have been directed towards developing efficient detectors [19, 57, 48, 34, 79, 75, 32, 31, 41]. Particularly, the YOLO series [48, 49, 50, 2, 20, 29, 62, 21, 65] stand out as the mainstream ones. YOLOv1, YOLOv2, and YOLOv3 identify the typical detection architecture consisting of three parts, *i.e.*, backbone, neck, and head [48, 49, 50]. YOLOv4 [2] and YOLOv5 [20] introduce the CSPNet [63] design to replace DarkNet [47], coupled with data augmentation strategies, enhanced PAN, and a greater variety of model scales, *etc*. YOLOv6 [29] presents BiC and SimCSPSPPF for neck and backbone, respectively, with anchor-aided training and self-distillation strategy. YOLOv7 [62] introduces E-ELAN for rich gradient flow path and explores several trainable bag-of-freebies methods. YOLOv8 [21] presents C2f building block for effective feature extraction and fusion. Gold-YOLO [60] provides the advanced GD mechanism to boost the multi-scale feature fusion capability. YOLOv9 [65] proposes GELAN to improve the architecture and PGI to augment the training process.

**End-to-end object detectors.** End-to-end object detection has emerged as a paradigm shift from traditional pipelines, offering streamlined architectures [53]. DETR [4] introduces the transformer architecture and adopts Hungarian loss to achieve one-to-one matching prediction, thereby eliminating hand-crafted components and post-processing. Since then, various DETR variants have been proposed to enhance its performance and efficiency [42, 67, 56, 30, 36, 28, 5, 77, 82]. Deformable-DETR [81] leverages multi-scale deformable attention module to accelerate the convergence speed. DINO [73] integrates contrastive denoising, mix query selection, and look forward twice scheme into DETRs. RT-DETR [78] further designs the efficient hybrid encoder and proposes the uncertainty-minimal query selection to improve both the accuracy and latency. Another line to achieve end-to-end object detection is based CNN detectors. Learnable NMS [24] and relation networks [26] present another network to remove duplicated predictions for detectors. OneNet [55] and DeFCN [66] propose one-to-one matching strategies to enable end-to-end object detection with fully convolutional networks. FCOS$_{pss}$ [80] introduces a positive sample selector to choose the optimal sample for prediction.

## 3 Methodology

### 3.1 Consistent Dual Assignments for NMS-free Training

During training, YOLOs [21, 65, 29, 70] usually leverage TAL [15] to allocate multiple positive samples for each instance. The adoption of one-to-many assignment yields plentiful supervisory signals, facilitating the optimization and achieving superior performance. However, it necessitates YOLOs to rely on the NMS post-processing, which causes the suboptimal inference efficiency for deployment. While previous works [55, 66, 80, 6] explore one-to-one matching to suppress the redundant predictions, they usually introduce additional inference overhead or yield suboptimal performance. In this work, we present a NMS-free training strategy for YOLOs with dual label assignments and consistent matching metric, achieving both high efficiency and competitive performance.

**Dual label assignments.** Unlike one-to-many assignment, one-to-one matching assigns only one prediction to each ground truth, avoiding the NMS post-processing. However, it leads to weak supervision, which causes suboptimal accuracy and convergence speed [82]. Fortunately, this deficiency can be compensated for by the one-to-many assignment [6]. To achieve this, we introduce

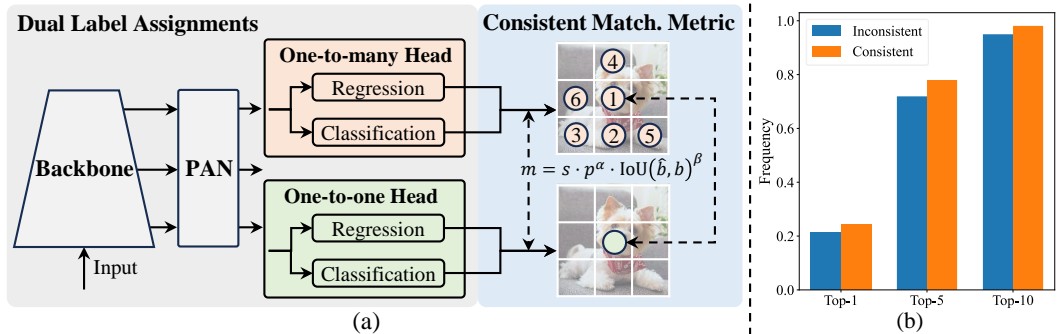

Figure 2: (a) Consistent dual assignments for NMS-free training. (b) Frequency of one-to-one assignments in Top-1/5/10 of one-to-many results for YOLOv8-S which employs $\alpha_{o2m}$=0.5 and $\beta_{o2m}$=6 by default [21]. For consistency, $\alpha_{o2o}$=0.5; $\beta_{o2o}$=6. For inconsistency, $\alpha_{o2o}$=0.5; $\beta_{o2o}$=2.

dual label assignments for YOLOs to combine the best of both strategies. Specifically, as shown in Fig. 2.(a), we incorporate another one-to-one head for YOLOs. It retains the identical structure and adopts the same optimization objectives as the original one-to-many branch but leverages the one-to-one matching to obtain label assignments. During training, two heads are jointly optimized with the model, allowing the backbone and neck to enjoy the rich supervision provided by the one-to-many assignment. During inference, we discard the one-to-many head and utilize the one-to-one head to make predictions. This enables YOLOs for the end-to-end deployment without incurring any additional inference cost. Besides, in the one-to-one matching, we adopt the top one selection, which achieves the same performance as Hungarian matching [4] with less extra training time.

**Consistent matching metric.** During assignments, both one-to-one and one-to-many approaches leverage a metric to quantitatively assess the level of concordance between predictions and instances. To achieve prediction aware matching for both branches, we employ a uniform matching metric, *i.e.*,

$$m(\alpha, \beta) = s \cdot p^\alpha \cdot \text{IoU}(\hat{b}, b)^\beta, \tag{1}$$

where $p$ is the classification score, $\hat{b}$ and $b$ denote the bounding box of prediction and instance, respectively. $s$ represents the spatial prior indicating whether the anchor point of prediction is within the instance [21, 65, 29, 70]. $\alpha$ and $\beta$ are two important hyperparameters that balance the impact of the semantic prediction task and the location regression task. We denote the one-to-many and one-to-one metrics as $m_{o2m}$=$m(\alpha_{o2m}, \beta_{o2m})$ and $m_{o2o}$=$m(\alpha_{o2o}, \beta_{o2o})$, respectively. These metrics influence the label assignments and supervision information for the two heads.

In dual label assignments, the one-to-many branch provides much richer supervisory signals than one-to-one branch. Intuitively, if we can harmonize the supervision of the one-to-one head with that of one-to-many head, we can optimize the one-to-one head towards the direction of one-to-many head's optimization. As a result, the one-to-one head can provide improved quality of samples during inference, leading to better performance. To this end, we first analyze the supervision gap between the two heads. Due to the randomness during training, we initiate our examination in the beginning with two heads initialized with the same values and producing the same predictions, *i.e.*, one-to-one head and one-to-many head generate the same $p$ and IoU for each prediction-instance pair. We note that the regression targets of two branches do not conflict, as matched predictions share the same targets and unmatched predictions are ignored. The supervision gap thus lies in the different classification targets. Given an instance, we denote its largest IoU with predictions as $u^*$, and the largest one-to-many and one-to-one matching scores as $m^*_{o2m}$ and $m^*_{o2o}$, respectively. Suppose that one-to-many branch yields the positive samples $\Omega$ and one-to-one branch selects $i$-th prediction with the metric $m_{o2o,i}$=$m^*_{o2o}$, we can then derive the classification target $t_{o2m,j}$=$u^* \cdot \frac{m_{o2m,j}}{m^*_{o2m}} \leq u^*$ for $j \in \Omega$ and $t_{o2o,i}$=$u^* \cdot \frac{m_{o2o,i}}{m^*_{o2o}}$=$u^*$ for task aligned loss as in [21, 65, 29, 70, 15]. The supervision gap between two branches can thus be derived by the 1-Wasserstein distance [46] of different classification objectives, *i.e.*,

$$A = t_{o2o,i} - \mathbb{I}(i \in \Omega)t_{o2m,i} + \sum\nolimits_{k \in \Omega \setminus \{i\}} t_{o2m,k}, \tag{2}$$

We can observe that the gap decreases as $t_{o2m,i}$ increases, *i.e.*, $i$ ranks higher within $\Omega$. It reaches the minimum when $t_{o2m,i}$=$u^*$, *i.e.*, $i$ is the best positive sample in $\Omega$, as shown in Fig. 2.(a). To achieve this, we present the consistent matching metric, *i.e.*, $\alpha_{o2o}$=$r \cdot \alpha_{o2m}$ and $\beta_{o2o}$=$r \cdot \beta_{o2m}$, which implies $m_{o2o}$=$m^r_{o2m}$. Therefore, the best positive sample for one-to-many head is also the best for one-to-one head. Consequently, both heads can be optimized consistently and harmoniously. For simplicity, we

take $r=1$, by default, *i.e.*, $\alpha_{o2o}=\alpha_{o2m}$ and $\beta_{o2o}=\beta_{o2m}$. To verify the improved supervision alignment, we count the number of one-to-one matching pairs within the top-1 / 5 / 10 of the one-to-many results after training. As shown in Fig. 2.(b), the alignment is improved under the consistent matching metric. For a more comprehensive understanding of the mathematical proof, please refer to the appendix.

**Discussion with other counter-parts.** Similarly, previous works [28, 5, 77, 54, 6, 82, 45] explore the different assignments to accelerate the training convergence and improve the performance for different networks. For example, H-DETR [28], Group-DETR [5], and MS-DETR [77] introduce one-to-many matching in conjunction with the original one-to-one matching by hybrid or multiple group label assignments, to improve upon DETR. Differently, to achieve the one-to-many matching, they usually introduce extra queries or repeat ground truths for bipartite matching, or select top several queries from the matching scores, while we adopt the prediction aware assignment that incorporates the spatial prior. Besides, LRANet [54] employs the dense assignment and sparse assignment branches for training, which all belong to the one-to-many assignment, while we adopt the one-to-many and one-to-one branches. DEYO [45, 43, 44] investigates the step-by-step training with one-to-many matching in the first stage for convolutional encoder and one-to-one matching in the second stage for transformer decoder, while we avoid the transformer decoder for end-to-end inference. Compared with works [6, 80] which incorporate dual assignments for CNN-based detectors, we further analyze the supervision gap between the two heads and present the consistent matching metric for YOLOs to reduce the theoretical supervision gap. It improves performance through better supervision alignment and eliminates the need for hyper-parameter tuning.

## 3.2  Holistic Efficiency-Accuracy Driven Model Design

In addition to the post-processing, the model architectures of YOLOs also pose great challenges to the efficiency-accuracy trade-offs [50, 8, 29]. Although previous works explore various design strategies, the comprehensive inspection for various components in YOLOs is still lacking. Consequently, the model architecture exhibits non-negligible computational redundancy and constrained capability, which impedes its potential for achieving high efficiency and performance. Here, we aim to holistically perform model designs for YOLOs from both efficiency and accuracy perspectives.

**Efficiency driven model design.** The components in YOLO consist of the stem, downsampling layers, stages with basic building blocks, and the head. The stem incurs few computational cost and we thus perform efficiency driven model design for other three parts.

*(1) Lightweight classification head.* The classification and regression heads usually share the same architecture in YOLOs. However, they exhibit notable disparities in computational overhead. For example, the FLOPs and parameter count of the classification head (5.95G/1.51M) are 2.5× and 2.4× those of the regression head (2.34G/0.64M) in YOLOv8-S, respectively. However, after analyzing the impact of classification error and the regression error (seeing Tab. 6), we find that the regression head undertakes more significance for the performance of YOLOs. Consequently, we can reduce the overhead of classification head without worrying about hurting the performance greatly. Therefore, we simply adopt a lightweight architecture for the classification head, which consists of two depthwise separable convolutions [25, 9] with the kernel size of 3×3 followed by a 1×1 convolution.

*(2) Spatial-channel decoupled downsampling.* YOLOs typically leverage regular 3×3 standard convolutions with stride of 2, achieving spatial downsampling (from $H \times W$ to $\frac{H}{2} \times \frac{W}{2}$) and channel transformation (from $C$ to $2C$) simultaneously. This introduces non-negligible computational cost of $\mathcal{O}(\frac{9}{2}HWC^2)$ and parameter count of $\mathcal{O}(18C^2)$. Instead, we propose to decouple the spatial reduction and channel increase operations, enabling more efficient downsampling. Specifically, we firstly leverage the pointwise convolution to modulate the channel dimension and then utilize the depthwise convolution to perform spatial downsampling. This reduces the computational cost to $\mathcal{O}(2HWC^2 + \frac{9}{2}HWC)$ and the parameter count to $\mathcal{O}(2C^2 + 18C)$. Meanwhile, it maximizes information retention during downsampling, leading to competitive performance with latency reduction.

*(3) Rank-guided block design.* YOLOs usually employ the same basic building block for all stages [29, 65], *e.g.*, the bottleneck block in YOLOv8 [21]. To thoroughly examine such homogeneous design for YOLOs, we utilize the intrinsic rank [33, 16] to analyze the redundancy[2] of each stage. Specifically, we calculate the numerical rank of the last convolution in the last basic block in each stage, which counts the number of singular values larger than a threshold. Fig. 3.(a) presents the results of

---

[2]A lower rank implies greater redundancy, while a higher rank signifies more condensed information.

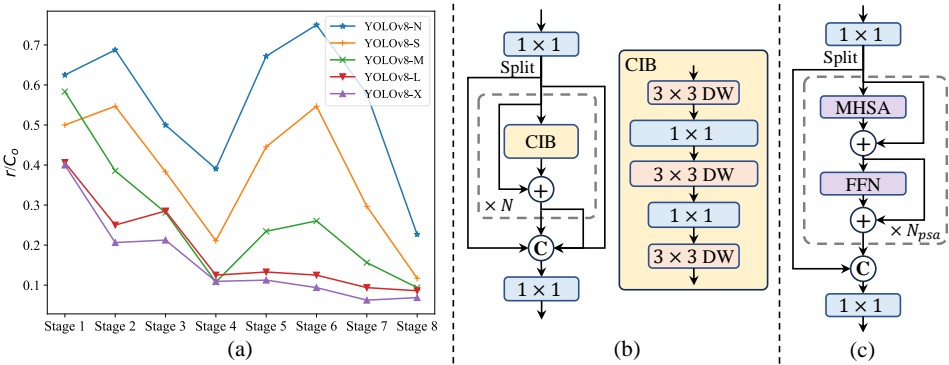

Figure 3: (a) The intrinsic ranks across stages and models in YOLOv8. The stage in the backbone and neck is numbered in the order of model forward process. The numerical rank $r$ is normalized to $r/C_o$ for y-axis and its threshold is set to $\lambda_{max}/2$, by default, where $C_o$ denotes the number of output channels and $\lambda_{max}$ is the largest singular value. It can be observed that deep stages and large models exhibit lower intrinsic rank values. (b) The compact inverted block (CIB). (c) The partial self-attention module (PSA).

YOLOv8, indicating that deep stages and large models are prone to exhibit more redundancy. This observation suggests that simply applying the same block design for all stages is suboptimal for the best capacity-efficiency trade-off. To tackle this, we propose a rank-guided block design scheme which aims to decrease the complexity of stages that are shown to be redundant using compact architecture design. We first present a compact inverted block (CIB) structure, which adopts the cheap depthwise convolutions for spatial mixing and cost-effective pointwise convolutions for channel mixing, as shown in Fig. 3.(b). It can serve as the efficient basic building block, *e.g.*, embedded in the ELAN structure [64, 21] (Fig. 3.(b)). Then, we advocate a rank-guided block allocation strategy to achieve the best efficiency while maintaining competitive capacity. Specifically, given a model, we sort its all stages based on their intrinsic ranks in ascending order. We further inspect the performance variation of replacing the basic block in the leading stage with CIB. If there is no performance degradation compared with the given model, we proceed with the replacement of the next stage and halt the process otherwise. Consequently, we can implement adaptive compact block designs across stages and model scales, achieving higher efficiency without compromising performance. Due to the page limit, we provide the details of the algorithm in the appendix.

**Accuracy driven model design.** We further explore the large-kernel convolution and self-attention for accuracy driven design, aiming to boost the performance under minimal cost.

*(1) Large-kernel convolution.* Employing large-kernel depthwise convolution is an effective way to enlarge the receptive field and enhance the model's capability [10, 40, 39]. However, simply leveraging them in all stages may introduce contamination in shallow features used for detecting small objects, while also introducing significant I/O overhead and latency in high-resolution stages [8]. Therefore, we propose to leverage the large-kernel depthwise convolutions in CIB within the deep stages. Specifically, we increase the kernel size of the second $3\times3$ depthwise convolution in the CIB to $7\times7$, following [39]. Additionally, we employ the structural reparameterization technique [11, 10, 59] to bring another $3\times3$ depthwise convolution branch to alleviate the optimization issue without inference overhead. Furthermore, as the model size increases, its receptive field naturally expands, with the benefit of using large-kernel convolutions diminishing. Therefore, we only adopt large-kernel convolution for small model scales.

*(2) Partial self-attention (PSA).* Self-attention [58] is widely employed in various visual tasks due to its remarkable global modeling capability [38, 14, 76]. However, it exhibits high computational complexity and memory footprint. To address this, in light of the prevalent attention head redundancy [69], we present an efficient partial self-attention (PSA) module design, as shown in Fig. 3.(c). Specifically, we evenly partition the features across channels into two parts after the $1\times1$ convolution. We only feed one part into the $N_{PSA}$ blocks comprised of multi-head self-attention module (MHSA) and feed-forward network (FFN). Two parts are then concatenated and fused by a $1\times1$ convolution. Besides, we follow [22] to assign the dimensions of the query and key to half of that of the value in MHSA and replace the `LayerNorm` [1] with `BatchNorm` [27] for fast inference. Furthermore, PSA is only placed after the Stage 4 with the lowest resolution, avoiding the excessive overhead from the

Table 1: Comparisons with state-of-the-arts. Latency is measured using official pre-trained models. Latency$^f$ denotes the latency in the forward process of model without post-processing. † means the results of YOLOv10 with the original one-to-many training using NMS. All results below are without the additional advanced training techniques like knowledge distillation or PGI for fair comparisons.

| Model | #Param.(M) | FLOPs(G) | AP$^{val}$(%) | Latency(ms) | Latency$^f$(ms) |
|---|---|---|---|---|---|
| YOLOv6-3.0-N [29] | 4.7 | 11.4 | 37.0 | 2.69 | 1.76 |
| Gold-YOLO-N [60] | 5.6 | 12.1 | 39.6 | 2.92 | 1.82 |
| YOLOv8-N [21] | 3.2 | 8.7 | 37.3 | 6.16 | 1.77 |
| **YOLOv10-N (Ours)** | **2.3** | **6.7** | **38.5 / 39.5**$^†$ | **1.84** | **1.79** |
| YOLOv6-3.0-S [29] | 18.5 | 45.3 | 44.3 | 3.42 | 2.35 |
| Gold-YOLO-S [60] | 21.5 | 46.0 | 45.4 | 3.82 | 2.73 |
| YOLO-MS-XS [8] | 4.5 | 17.4 | 43.4 | 8.23 | 2.80 |
| YOLO-MS-S [8] | 8.1 | 31.2 | 46.2 | 10.12 | 4.83 |
| YOLOv8-S [21] | 11.2 | 28.6 | 44.9 | 7.07 | 2.33 |
| YOLOv9-S [65] | 7.1 | 26.4 | 46.7 | - | - |
| RT-DETR-R18 [78] | 20.0 | 60.0 | 46.5 | 4.58 | 4.49 |
| **YOLOv10-S (Ours)** | **7.2** | **21.6** | **46.3 / 46.8**$^†$ | **2.49** | **2.39** |
| YOLOv6-3.0-M [29] | 34.9 | 85.8 | 49.1 | 5.63 | 4.56 |
| Gold-YOLO-M [60] | 41.3 | 87.5 | 49.8 | 6.38 | 5.45 |
| YOLO-MS [8] | 22.2 | 80.2 | 51.0 | 12.41 | 7.30 |
| YOLOv8-M [21] | 25.9 | 78.9 | 50.6 | 9.50 | 5.09 |
| YOLOv9-M [65] | 20.0 | 76.3 | 51.1 | - | - |
| RT-DETR-R34 [78] | 31.0 | 92.0 | 48.9 | 6.32 | 6.21 |
| RT-DETR-R50m [78] | 36.0 | 100.0 | 51.3 | 6.90 | 6.84 |
| **YOLOv10-M (Ours)** | **15.4** | **59.1** | **51.1 / 51.3**$^†$ | **4.74** | **4.63** |
| YOLOv6-3.0-L [29] | 59.6 | 150.7 | 51.8 | 9.02 | 7.90 |
| Gold-YOLO-L [60] | 75.1 | 151.7 | 51.8 | 10.65 | 9.78 |
| YOLOv9-C [65] | 25.3 | 102.1 | 52.5 | 10.57 | 6.13 |
| **YOLOv10-B (Ours)** | **19.1** | **92.0** | **52.5 / 52.7**$^†$ | **5.74** | **5.67** |
| YOLOv8-L [21] | 43.7 | 165.2 | 52.9 | 12.39 | 8.06 |
| RT-DETR-R50 [78] | 42.0 | 136.0 | 53.1 | 9.20 | 9.07 |
| **YOLOv10-L (Ours)** | **24.4** | **120.3** | **53.2 / 53.4**$^†$ | **7.28** | **7.21** |
| YOLOv8-X [21] | 68.2 | 257.8 | 53.9 | 16.86 | 12.83 |
| RT-DETR-R101 [78] | 76.0 | 259.0 | 54.3 | 13.71 | 13.58 |
| **YOLOv10-X (Ours)** | **29.5** | **160.4** | **54.4 / 54.4**$^†$ | **10.70** | **10.60** |

quadratic computational complexity of self-attention. In this way, the global representation learning ability can be incorporated into YOLOs with low computational costs, which well enhances the model's capability and leads to improved performance.

# 4 Experiments

## 4.1 Implementation Details

We select YOLOv8 [21] as our baseline model, due to its commendable latency-accuracy balance and its availability in various model sizes. We employ the consistent dual assignments for NMS-free training and perform holistic efficiency-accuracy driven model design based on it, which brings our YOLOv10 models. YOLOv10 has the same variants as YOLOv8, *i.e.*, N / S / M / L / X. Besides, we derive a new variant YOLOv10-B, by simply increasing the width scale factor of YOLOv10-M. We verify the proposed detector on COCO [35] under the same training-from-scratch setting [21, 65, 62]. Moreover, the latencies of all models are tested on T4 GPU with TensorRT FP16, following [78].

## 4.2 Comparison with state-of-the-arts

As shown in Tab. 1, our YOLOv10 achieves the state-of-the-art performance and end-to-end latency across various model scales. We first compare YOLOv10 with our baseline models, *i.e.*, YOLOv8. On N / S / M / L / X five variants, our YOLOv10 achieves 1.2% / 1.4% / 0.5% / 0.3% / 0.5% AP improvements, with 28% / 36% / 41% / 44% / 57% fewer parameters, 23% / 24% / 25% / 27% / 38% less calculations, and 70% / 65% / 50% / 41% / 37% lower latencies. Compared with other YOLOs,

Table 2: Ablation study with YOLOv10-S and YOLOv10-M on COCO.

| # | Model | NMS-free. | Efficiency. | Accuracy. | #Param.(M) | FLOPs(G) | $AP^{val}$(%) | Latency(ms) |
|---|---|---|---|---|---|---|---|---|
| 1 | YOLOv10-S | | | | 11.2 | 28.6 | 44.9 | 7.07 |
| 2 | | ✓ | | | 11.2 | 28.6 | 44.3 | 2.44 |
| 3 | | ✓ | ✓ | | 6.2 | 20.8 | 44.5 | 2.31 |
| 4 | | ✓ | ✓ | ✓ | 7.2 | 21.6 | 46.3 | 2.49 |
| 5 | YOLOv10-M | | | | 25.9 | 78.9 | 50.6 | 9.50 |
| 6 | | ✓ | | | 25.9 | 78.9 | 50.3 | 5.22 |
| 7 | | ✓ | ✓ | | 14.1 | 58.1 | 50.4 | 4.57 |
| 8 | | ✓ | ✓ | ✓ | 15.4 | 59.1 | 51.1 | 4.74 |

Table 3: Dual assign.

| o2m | o2o | AP | Latency |
|---|---|---|---|
| ✓ | | 44.9 | 7.07 |
| | ✓ | 43.4 | 2.44 |
| ✓ | ✓ | 44.3 | 2.44 |

Table 4: Matching metric.

| $\alpha_{o2o}$ | $\beta_{o2o}$ | $AP^{val}$ | $\alpha_{o2o}$ | $\beta_{o2o}$ | $AP^{val}$ |
|---|---|---|---|---|---|
| 0.5 | 2.0 | 42.7 | 0.25 | 3.0 | 44.3 |
| 0.5 | 4.0 | 44.2 | 0.25 | 6.0 | 43.5 |
| 0.5 | 6.0 | 44.3 | 1.0 | 6.0 | 43.9 |
| 0.5 | 8.0 | 44.0 | 1.0 | 12.0 | 44.3 |

Table 5: Efficiency. for YOLOv10-S/M.

| # | Model | #Param | FLOPs | $AP^{val}$ | Latency |
|---|---|---|---|---|---|
| 1 | base. | 11.2/25.9 | 28.6/78.9 | 44.3/50.3 | 2.44/5.22 |
| 2 | +cls. | 9.9/23.2 | 23.5/67.7 | 44.2/50.2 | 2.39/5.07 |
| 3 | +downs. | 8.0/19.7 | 22.2/65.0 | 44.4/50.4 | 2.36/4.97 |
| 4 | +block. | 6.2/14.1 | 20.8/58.1 | 44.5/50.4 | 2.31/4.57 |

YOLOv10 also exhibits superior trade-offs between accuracy and computational cost. Specifically, for lightweight and small models, YOLOv10-N / S outperforms YOLOv6-3.0-N / S by 1.5 AP and 2.0 AP, with 51% / 61% fewer parameters and 41% / 52% less computations, respectively. For medium models, compared with YOLOv9-C / YOLO-MS, YOLOv10-B / M enjoys the 46% / 62% latency reduction under the same or better performance, respectively. For large models, compared with Gold-YOLO-L, our YOLOv10-L shows 68% fewer parameters and 32% lower latency, along with a significant improvement of 1.4% AP. Furthermore, compared with RT-DETR, YOLOv10 obtains significant performance and latency improvements. Notably, YOLOv10-S / X achieves 1.8× and 1.3× faster inference speed than RT-DETR-R18 / R101, respectively, under the similar performance. These results well demonstrate the superiority of YOLOv10 as the real-time end-to-end detector.

We also compare YOLOv10 with other YOLOs using the original one-to-many training approach. We consider the performance and the latency of model forward process (Latency$^f$) in this situation, following [62, 21, 60]. As shown in Tab. 1, YOLOv10 also exhibits the state-of-the-art performance and efficiency across different model scales, indicating the effectiveness of our architectural designs.

### 4.3 Model Analyses

**Ablation study.** We present the ablation results based on YOLOv10-S and YOLOv10-M in Tab. 2. It can be observed that our NMS-free training with consistent dual assignments significantly reduces the end-to-end latency of YOLOv10-S by 4.63ms, while maintaining competitive performance of 44.3% AP. Moreover, our efficiency driven model design leads to the reduction of 11.8 M parameters and 20.8 GFlOPs, with a considerable latency reduction of 0.65ms for YOLOv10-M, well showing its effectiveness. Furthermore, our accuracy driven model design achieves the notable improvements of 1.8 AP and 0.7 AP for YOLOv10-S and YOLOv10-M, alone with only 0.18ms and 0.17ms latency overhead, respectively, which well demonstrates its superiority.

**Analyses for NMS-free training.**
- *Dual label assignments.* We present dual label assignments for NMS-free YOLOs, which can bring both rich supervision of one-to-many (o2m) branch during training and high efficiency of one-to-one (o2o) branch during inference. We verify its benefit based on YOLOv8-S, *i.e.*, #1 in Tab. 2. Specifically, we introduce baselines for training with only o2m branch and only o2o branch, respectively. As shown in Tab. 3, our dual label assignments achieve the best AP-latency trade-off.
- *Consistent matching metric.* We introduce consistent matching metric to make the one-to-one head more harmonious with the one-to-many head. We verify its benefit based on YOLOv8-S, *i.e.*, #1 in Tab. 2, under different $\alpha_{o2o}$ and $\beta_{o2o}$. As shown in Tab. 4, the proposed consistent matching metric, *i.e.*, $\alpha_{o2o}=r \cdot \alpha_{o2m}$ and $\beta_{o2o}=r \cdot \beta_{o2m}$, can achieve the optimal performance, where $\alpha_{o2m}=0.5$ and $\beta_{o2m}=6.0$ in the one-to-many head [21]. Such an improvement can be attributed to the reduction of the supervision gap (Eq. (2)), which provides improved supervision alignment between two branches. Moreover, the proposed consistent matching metric eliminates the need for exhaustive hyper-parameter tuning, which is appealing in practical scenarios.

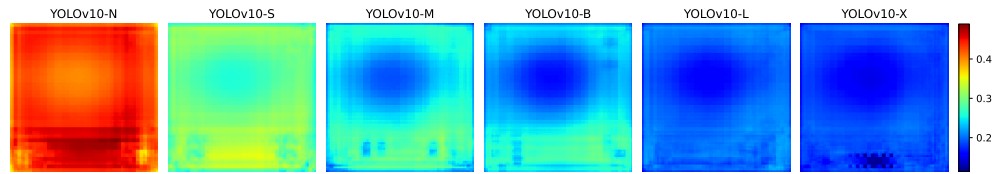

Figure 4: The average cosine similarity of each anchor point's extracted features with all others.

Table 6: cls. results.

|  | base. | +cls. |
|---|---|---|
| $AP^{val}$ | 44.3 | 44.2 |
| $AP^{val}_{w/o\ c}$ | 59.9 | 59.9 |
| $AP^{val}_{w/o\ r}$ | 64.5 | 64.2 |

Table 7: Results of d.s.

| Model | $AP^{val}$ | Latency |
|---|---|---|
| base. | 43.7 | 2.33 |
| ours | 44.4 | 2.36 |

Table 8: Results of CIB.

| Model | $AP^{val}$ | Latency |
|---|---|---|
| IRB | 43.7 | 2.30 |
| IRB-DW | 44.2 | 2.30 |
| ours | 44.5 | 2.31 |

Table 9: Rank-guided.

| Stages with CIB | $AP^{val}$ |
|---|---|
| empty | 44.4 |
| 8 | 44.5 |
| 8,4, | 44.5 |
| 8,4,7 | 44.3 |

- *Performance gap compared with one-to-many training.* Although achieving superior end-to-end performance under NMS-free training, we observe that there still exists the performance gap compared with the original one-to-many training using NMS, as shown in Tab. 3 and Tab. 1. Besides, we note that the gap diminishes as the model size increases. Therefore, we reasonably concludes that such a gap can be attributed to the limitations in the model capability. Notably, unlike the original one-to-many training using NMS, the NMS-free training necessitates more discriminative features for one-to-one matching. In the case of the YOLOv10-N model, its limited capacity results in extracted features that lack sufficient discriminability, leading to a more notable performance gap of 1.0% AP. In contrast, the YOLOv10-X model, which possesses stronger capability and more discriminative features, shows no performance gap between two training strategies. In Fig. 4, we visualize the average cosine similarity of each anchor point's extracted features with those of all other anchor points on the COCO `val` set. We observe that as the model size increases, the feature similarity between anchor points exhibits a downward trend, which benefits the one-to-one matching. Based on this insight, we will explore approaches to further reduce the gap and achieve higher end-to-end performance in the future work.

**Analyses for efficiency driven model design**. We conduct experiments to gradually incorporate the efficiency driven design elements based on YOLOv10-S/M. Our baseline is the YOLOv10-S/M model without efficiency-accuracy driven model design, *i.e.*, #2/#6 in Tab. 2. As shown in Tab. 5, each design component, including lightweight classification head, spatial-channel decoupled downsampling, and rank-guided block design, contributes to the reduction of parameters count, FLOPs, and latency. Importantly, these improvements are achieved while maintaining competitive performance.

- *Lightweight classification head.* We analyze the impact of category and localization errors of predictions on the performance, based on the YOLOv10-S of #1 and #2 in Tab. 5, like [7]. Specifically, we match the predictions to the instances by the one-to-one assignment. Then, we substitute the predicted category score with instance labels, resulting in $AP^{val}_{w/o\ c}$ with no classification errors. Similarly, we replace the predicted locations with those of instances, yielding $AP^{val}_{w/o\ r}$ with no regression errors. As shown in Tab. 6, $AP^{val}_{w/o\ r}$ is much higher than $AP^{val}_{w/o\ c}$, revealing that eliminating the regression errors achieves greater improvement. The performance bottleneck thus lies more in the regression task. Therefore, adopting the lightweight classification head can allow higher efficiency without compromising the performance.

- *Spatial-channel decoupled downsampling.* We decouple the downsampling operations for efficiency, where the channel dimensions are first increased by pointwise convolution (PW) and the resolution is then reduced by depthwise convolution (DW) for maximal information retention. We compare it with the baseline way of spatial reduction by DW followed by channel modulation by PW, based on the YOLOv10-S of #3 in Tab. 5. As shown in Tab. 7, our downsampling strategy achieves the 0.7% AP improvement by enjoying less information loss during downsampling.

- *Compact inverted block (CIB).* We introduce CIB as the compact basic building block. We verify its effectiveness based on the YOLOv10-S of #4 in the Tab. 5. Specifically, we introduce the inverted residual block [51] (IRB) as the baseline, which achieves the suboptimal 43.7% AP, as shown in Tab. 8. We then append a 3×3 depthwise convolution (DW) after it, denoted as "IRB-DW", which

Table 10: Accuracy. for S/M.  Table 11: L.k. results. Table 12: L.k. usage. Table 13: PSA results.

| # | Model | $AP^{val}$ | Latency |
|---|-------|-----------|---------|
| 1 | base. | 44.5/50.4 | 2.31/4.57 |
| 2 | +L.k. | 44.9/- | 2.34/- |
| 3 | +PSA | 46.3/51.1 | 2.49/4.74 |

| Model | $AP^{val}$ | Latency |
|-------|-----------|---------|
| k.s.=5 | 44.7 | 2.32 |
| k.s.=7 | 44.9 | 2.34 |
| k.s.=9 | 44.9 | 2.37 |
| w/o rep. | 44.8 | 2.34 |

| | w/o L.k. | w/ L.k. |
|---|---------|---------|
| N | 36.3 | 36.6 |
| S | 44.5 | 44.9 |
| M | 50.4 | 50.4 |

| Model | $AP^{val}$ | Latency |
|-------|-----------|---------|
| PSA | 46.3 | 2.49 |
| Trans. | 46.0 | 2.54 |
| $N_{PSA} = 1$ | 46.3 | 2.49 |
| $N_{PSA} = 2$ | 46.5 | 2.59 |

brings 0.5% AP improvement. Compared with "IRB-DW", our CIB further achieves 0.3% AP improvement by prepending another DW with minimal overhead, indicating its superiority.

- *Rank-guided block design.* We introduce the rank-guided block design to adaptively integrate compact block design for improving the model efficiency. We verify its benefit based on the YOLOv10-S of #3 in the Tab. 5. The stages sorted in ascending order based on the intrinsic ranks are Stage 8-4-7-3-5-1-6-2, like in Fig. 3.(a). As shown in Tab. 9, when gradually replacing the bottleneck block in each stage with the efficient CIB, we observe the performance degradation starting from Stage 7. In the Stage 8 and 4 with lower intrinsic ranks and more redundancy, we can thus adopt the efficient block design without compromising the performance. These results indicate that rank-guided block design can serve as an effective strategy for higher model efficiency.

**Analyses for accuracy driven model design.** We present the results of gradually integrating the accuracy driven design elements based on YOLOv10-S/M. Our baseline is the YOLOv10-S/M model after incorporating efficiency driven design, *i.e.*, #3/#7 in Tab. 2. As shown in Tab. 10, the adoption of large-kernel convolution and PSA module leads to the considerable performance improvements of 0.4% AP and 1.4% AP for YOLOv10-S under minimal latency increase of 0.03ms and 0.15ms, respectively. Note that large-kernel convolution is not employed for YOLOv10-M (see Tab. 12).

- *Large-kernel convolution.* We first investigate the effect of different kernel sizes based on the YOLOv10-S of #2 in Tab. 10. As shown in Tab. 11, the performance improves as the kernel size increases and stagnates around the kernel size of $7 \times 7$, indicating the benefit of large perception field. Besides, removing the reparameterization branch during training achieves 0.1% AP degradation, showing its effectiveness for optimization. Moreover, we inspect the benefit of large-kernel convolution across model scales based on YOLOv10-N / S / M. As shown in Tab. 12 brings no improvements for large models, *i.e.*, YOLOv10-M, due to its inherent extensive receptive field. We thus only adopt large-kernel convolutions for small models, *i.e.*, YOLOv10-N / S.

- *Partial self-attention (PSA).* We introduce PSA to enhance the performance by incorporating the global modeling ability under minimal cost. We first verify its effectiveness based on the YOLOv10-S of #3 in Tab. 10. Specifically, we introduce the transformer block, *i.e.*, MHSA followed by FFN, as the baseline, denoted as "Trans.". As shown in Tab. 13, compared with it, PSA brings 0.3% AP improvement with 0.05ms latency reduction. The performance enhancement may be attributed to the alleviation of optimization problem [68, 10] in self-attention, by mitigating the redundancy in attention heads. Moreover, we investigate the impact of different $N_{PSA}$. As shown in Tab. 13, increasing $N_{PSA}$ to 2 obtains 0.2% AP improvement but with 0.1ms latency overhead. Therefore, we set $N_{PSA}$ to 1, by default, to enhance the model capability while maintaining high efficiency.

# 5 Conclusion

In this paper, we target both the post-processing and model architecture throughout the detection pipeline of YOLOs. For the post-processing, we propose the consistent dual assignments for NMS-free training, achieving efficient end-to-end detection. For the model architecture, we introduce the holistic efficiency-accuracy driven model design strategy, improving the performance-efficiency trade-offs. These bring our YOLOv10, a new real-time end-to-end object detector. Extensive experiments show that YOLOv10 achieves the state-of-the-art performance and latency compared with other advanced detectors, well demonstrating its superiority.

# 6 Acknowledgments

This work was supported by National Natural Science Foundation of China (Nos. 61925107, 62271281) and Beijing Natural Science Foundation (No. L223023).

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

# A Appendix

## A.1 Implementation Details

Following [21, 62, 65], all YOLOv10 models are trained from scratch using the SGD optimizer for 500 epochs. The SGD momentum and weight decay are set to 0.937 and $5\times10^{-4}$, respectively. The initial learning rate is $1\times10^{-2}$ and it decays linearly to $1\times10^{-4}$. For data augmentation, we adopt the Mosaic [2, 20], Mixup [74] and copy-paste augmentation [18], *etc.*, like [21, 65]. Tab. 14 presents the detailed hyper-parameters. All models are trained on 8 NVIDIA 3090 GPUs. Besides, we increase the width scale factor of YOLOv10-M to 1.0 to obtain YOLOv10-B. For PSA, we employ it after the SPPF module [21] and adopt the expansion factor of 2 for FFN. For CIB, we also adopt the expansion ratio of 2 for the inverted bottleneck block structure. Following [65, 62], we report the standard mean average precision (AP) across different object scales and IoU thresholds on the COCO dataset [35].

Moreover, we follow [78] to establish the end-to-end speed benchmark. Since the execution time of NMS is affected by the input, we thus measure the latency on the COCO `val` set with the batch size of 1, like [78]. We adopt the same NMS hyperparameters used by the detectors during their validation. The TensorRT `efficientNMSPlugin` is appended for post-processing and the I/O overhead is omitted. We report the average latency across all images.

Table 14: Hyper-parameters of YOLOv10.

| hyper-parameter | YOLOv10-N/S/M/B/L/X |
|---|---|
| epochs | 500 |
| optimizer | SGD |
| momentum | 0.937 |
| weight decay | $5\times10^{-4}$ |
| warm-up epochs | 3 |
| warm-up momentum | 0.8 |
| warm-up bias learning rate | 0.1 |
| initial learning rate | $10^{-2}$ |
| final learning rate | $10^{-4}$ |
| learning rate schedule | linear decay |
| box loss gain | 7.5 |
| class loss gain | 0.5 |
| DFL loss gain | 1.5 |
| HSV saturation augmentation | 0.7 |
| HSV value augmentation | 0.4 |
| HSV hue augmentation | 0.015 |
| translation augmentation | 0.1 |
| scale augmentation | 0.5/0.5/0.9/0.9/0.9/0.9 |
| mosaic augmentation | 1.0 |
| Mixup augmentation | 0.0/0.0/0.1/0.1/0.15/0.15 |
| copy-paste augmentation | 0.0/0.0/0.1/0.1/0.3/0.3 |
| close mosaic epochs | 10 |

## A.2 Details of Consistent Matching Metric

We provide the detailed derivation of consistent matching metric here.

As mentioned in the paper, we suppose that the one-to-many positive samples is $\Omega$ and the one-to-one branch selects $i$-th prediction. We can then leverage the normalized metric [15] to obtain the classification target for task alignment learning [21, 15, 65, 29, 70], *i.e.*, $t_{o2m,j} = u^* \cdot \frac{m_{o2m,j}}{m^*_{o2m}} \leq u^*$ for $j \in \Omega$ and $t_{o2o,i} = u^* \cdot \frac{m_{o2o,i}}{m^*_{o2o}} = u^*$. We can thus derive the supervision gap between two branches by the 1-Wasserstein distance [46] of the different classification targets, *i.e.*,

$$A = |(1 - t_{o2o,i}) - (1 - \mathbb{I}(i \in \Omega)t_{o2m,i})| + \sum_{k \in \Omega \setminus \{i\}} |1 - (1 - t_{o2m,k})|$$

$$= |t_{o2o,i} - \mathbb{I}(i \in \Omega)t_{o2m,i}| + \sum_{k \in \Omega \setminus \{i\}} t_{o2m,k} \quad (3)$$

$$= t_{o2o,i} - \mathbb{I}(i \in \Omega)t_{o2m,i} + \sum_{k \in \Omega \setminus \{i\}} t_{o2m,k},$$

where $\mathbb{I}(\cdot)$ is the indicator function. We denote the classification targets of the predictions in $\Omega$ as $\{\hat{t}_1, \hat{t}_2, ..., \hat{t}_{|\Omega|}\}$ in descending order, with $\hat{t}_1 \geq \hat{t}_2 \geq ... \geq \hat{t}_{|\Omega|}$. We can then replace $t_{o2o,i}$ with $u^*$ and obtain:

$$A = u^* - \mathbb{I}(i \in \Omega)t_{o2m,i} + \sum\nolimits_{k \in \Omega \backslash \{i\}} t_{o2m,k}$$
$$= u^* + \sum\nolimits_{k \in \Omega} t_{o2m,k} - 2 \cdot \mathbb{I}(i \in \Omega)t_{o2m,i} \qquad (4)$$
$$= u^* + \sum\nolimits_{k=1}^{|\Omega|} \hat{t}_k - 2 \cdot \mathbb{I}(i \in \Omega)t_{o2m,i}$$

We further discuss the supervision gap in two scenarios, *i.e.*,

1. Supposing $i \notin \Omega$, we can obtain:

$$A = u^* + \sum\nolimits_{k=1}^{|\Omega|} \hat{t}_k \qquad (5)$$

2. Supposing $i \in \Omega$, we denote $t_{o2m,i} = \hat{t}_n$ and obtain:

$$A = u^* + \sum\nolimits_{k=1}^{|\Omega|} \hat{t}_k - 2 \cdot \hat{t}_n \qquad (6)$$

Due to $\hat{t}_n \geq 0$, the second case can lead to smaller supervision gap. Besides, we can observe that $A$ decreases as $\hat{t}_n$ increases, indicating that $n$ decreases and the ranking of $i$ within $\Omega$ improves. Due to $\hat{t}_n \leq \hat{t}_1$, $A$ thus achieves the minimum when $\hat{t}_n = \hat{t}_1$, *i.e.*, $i$ is the best positive sample in $\Omega$ with $m_{o2m,i} = m^*_{o2m}$ and $t_{o2m,i} = u^* \cdot \frac{m_{o2m,i}}{m^*_{o2m}} = u^*$.

Furthermore, we prove that we can achieve the minimized supervision gap by the consistent matching metric. We suppose $\alpha_{o2m} > 0$ and $\beta_{o2m} > 0$, which are common in [21, 65, 29, 15, 70]. Similarly, we assume $\alpha_{o2o} > 0$ and $\beta_{o2o} > 0$. We can obtain $r_1 = \frac{\alpha_{o2o}}{\alpha_{o2m}} > 0$ and $r_2 = \frac{\beta_{o2o}}{\beta_{o2m}} > 0$, and then derive $m_{o2o}$ by

$$m_{o2o} = s \cdot p^{\alpha_{o2o}} \cdot \text{IoU}(\hat{b}, b)^{\beta_{o2o}}$$
$$= s \cdot p^{r_1 \cdot \alpha_{o2m}} \cdot \text{IoU}(\hat{b}, b)^{r_2 \cdot \beta_{o2m}} \qquad (7)$$
$$= s \cdot (p^{\alpha_{o2m}} \cdot \text{IoU}(\hat{b}, b)^{\beta_{o2m}})^{r_1} \cdot \text{IoU}(\hat{b}, b)^{(r_2 - r_1) \cdot \beta_{o2m}}$$
$$= m^{r_1}_{o2m} \cdot \text{IoU}(\hat{b}, b)^{(r_2 - r_1) \cdot \beta_{o2m}}$$

To achieve $m_{o2m,i} = m^*_{o2m}$ and $m_{o2o,i} = m^*_{o2o}$, we can make $m_{o2o}$ monotonically increase with $m_{o2m}$ by assigning $(r_2 - r_1) = 0$, *i.e.*,

$$m_{o2o} = m^{r_1}_{o2m} \cdot \text{IoU}(\hat{b}, b)^{0 \cdot \beta_{o2m}}$$
$$= m^{r_1}_{o2m} \qquad (8)$$

Supposing $r_1 = r_2 = r$, we can thus derive the consistent matching metric, *i.e.*, $\alpha_{o2o} = r \cdot \alpha_{o2m}$ and $\beta_{o2o} = r \cdot \beta_{o2m}$. By simply taking $r = 1$, we obtain $\alpha_{o2o} = \alpha_{o2m}$ and $\beta_{o2o} = \beta_{o2m}$.

### A.3 Details of Rank-Guided Block Design

We present the details of the algorithm of rank-guided block design in Algo. 1. Besides, to calculate the numerical rank of the convolution, we reshape its weight to the shape of $(C_o, K^2 \times C_i)$, where $C_o$ and $C_i$ denote the number of output and input channels, and $K$ means the kernel size, respectively.

### A.4 Training Cost Analyses

In addition to the inference efficiency analyses, we also investigate the training cost of our YOLOv10 models. We compare with other YOLO variants and measure the training throughput on 8 NVIDIA 3090 GPUs using the official codebases. Tab. 15 presents the comparison results based on the medium model scale. We observe that despite having 500 training epochs, YOLOv10 achieves a high training throughput, making its training cost affordable. We also note that the one-to-many head in the NMS-free training will introduce the extra overhead for YOLOv10. To investigate this, we measure the training cost of YOLOv10 with only the one-to-one head, which is denoted as "YOLOv10-o2o". As shown in Tab. 15, YOLOv10-M results in a small increase in the training time over "YOLOv10-M-o2o", about 18s each epoch, which is affordable. To fairly verify the benefit of the one-to-many head in NMS-free training, we also adopt longer 550 training epochs for "YOLOv10-M-o2o", which

**Algorithm 1:** Rank-guided block design

---

**Input:** Intrinsic ranks $R$ for all stages $S$; Original Network $\Theta$; CIB $\theta_{cib}$;
**Output:** New network $\Theta^*$ with CIB for certain stages.

1   $t \leftarrow 0$;
2   $\Theta_0 \leftarrow \Theta$; $\Theta^* \leftarrow \Theta_0$;
3   $ap_0 \leftarrow \text{AP}(\text{T}(\Theta_0))$;     // T:training the network; AP:evaluating the AP performance.
4   **while** $S \neq \emptyset$ **do**
5      $s_t \leftarrow \text{argmin}_{s \in S}\, R$;
6      $\Theta_{t+1} \leftarrow \text{Replace}(\Theta_t, \theta_{cib}, s_t)$; // Replace the block in Stage $s_t$ of $\Theta_t$ with CIB $\theta_{cib}$.
7      $ap_{t+1} \leftarrow \text{AP}(\text{T}(\Theta_{t+1}))$;
8      **if** $ap_{t+1} \geq ap_0$ **then**
9         $\Theta^* \leftarrow \Theta_{t+1}$; $S \leftarrow S \setminus \{s_t\}$;
10     **else**
11        **return** $\Theta^*$;
12     **end**
13 **end**
14 **return** $\Theta^*$;

---

Table 15: Training cost analyses on 8 NVIDIA 3090 GPUs.

| Model | Epoch | Speed (epoch/hour) | Time (hour) |
|---|---|---|---|
| YOLOv6-3.0-M | 300 | 7.2 | 41.7 |
| YOLOv8-M | 500 | 18.3 | 27.3 |
| YOLOv9-M | 500 | 12.3 | 40.7 |
| Gold-YOLO-M | 300 | 4.7 | 63.8 |
| YOLO-MS | 300 | 7.1 | 42.3 |
| YOLOv10-M-o2o | 500 | 18.8 | 26.7 |
| **YOLOv10-M** | 500 | **17.2** | **29.1** |

Table 16: Latency with NMS.

| Model | Latency |
|---|---|
| YOLOv10-N | 6.19ms |
| YOLOv10-S | 7.15ms |
| YOLOv10-M | 9.03ms |
| YOLOv10-B | 10.04ms |
| YOLOv10-L | 11.52ms |
| YOLOv10-X | 14.67ms |

leads to a similar training time (29.3 vs. 29.1 hours) but still yields inferior performance (48.9% vs. 51.1% AP) compared with YOLOv10-M.

### A.5   More Results on COCO

We measure the latency of YOLOv10 with the original one-to-many training using NMS and report the results on COCO in Tab. 16. Besides, we report the detailed performance of YOLOv10, including $AP_{50}^{val}$ and $AP_{75}^{val}$ at different IoU thresholds, as well as $AP_{small}^{val}$, $AP_{medium}^{val}$, and $AP_{large}^{val}$ across different scales, in Tab. 17. We also present the comparisons with more lightweight detectors, including DAMO-YOLO [71], YOLOv7 [62], and DEYO [45], in Tab. 18. It shows that our YOLOv10 also achieves superior performance and efficiency trade-offs. Additionally, in experiments, we follow previous works [21, 65] to train the models for 500 epochs. We also conduct experiments to train the models for 300 epochs and present the comparison results with YOLOv6 [29], Gold-YOLO [60], and YOLO-MS [8] which adopt 300 epochs, in Tab. 19. We observe that our YOLOv10 also exhibits better performance and inference latency. We also note that despite trained for 500 epochs, YOLOv10 has less training cost compared with these models as presented in Tab. 15.

### A.6   Inference Efficiency Comparison on CPU

We present the speed comparison results of YOLOv10 and others on CPU (Intel Xeon Skylake, IBRS) using OpenVINO in Fig. 5. We observe that YOLOv10 also shows state-of-the-art trade-offs in terms of performance and efficiency.

### A.7   More Analyses for Holistic Efficiency-Accuracy Driven Model Design

We note that reducing the latency of YOLOv10-S (#2 in Tab. 2) is particularly challenging due to its small model scale. However, as shown in Tab. 2, our efficiency driven model design still achieves a 5.3% reduction in latency without compromising performance. This provides substantial support for the further accuracy driven model design. YOLOv10-S achieves a better latency-accuracy trade-off

Table 17: Detailed performance of YOLOv10 on COCO.

| Model | $AP^{val}$(%) | $AP_{50}^{val}$(%) | $AP_{75}^{val}$(%) | $AP_{small}^{val}$(%) | $AP_{medium}^{val}$(%) | $AP_{large}^{val}$(%) |
|---|---|---|---|---|---|---|
| YOLOv10-N | 38.5 | 53.8 | 41.7 | 18.9 | 42.4 | 54.6 |
| YOLOv10-S | 46.3 | 63.0 | 50.4 | 26.8 | 51.0 | 63.8 |
| YOLOv10-M | 51.1 | 68.1 | 55.8 | 33.8 | 56.5 | 67.0 |
| YOLOv10-B | 52.5 | 69.6 | 57.2 | 35.1 | 57.8 | 68.5 |
| YOLOv10-L | 53.2 | 70.1 | 58.1 | 35.8 | 58.5 | 69.4 |
| YOLOv10-X | 54.4 | 71.3 | 59.3 | 37.0 | 59.8 | 70.9 |

Table 18: Comparisons with more lightweight detectors.

| Model | #Param.(M) | FLOPs(G) | $AP^{val}$(%) | Latency(ms) |
|---|---|---|---|---|
| DEYO-tiny [45] | 4.0 | 8.0 | 37.6 | 2.01 |
| **YOLOv10-N** | **2.3** | **6.7** | **38.5** | **1.84** |
| DAMO-YOLO-T [71] | 8.5 | 18.1 | 42.0 | 2.21 |
| DAMO-YOLO-S [71] | 16.3 | 37.8 | 46.0 | 3.18 |
| DEYO-S [45] | 14.0 | 26.0 | 45.8 | 3.34 |
| **YOLOv10-S** | **7.2** | **21.6** | **46.3** | **2.49** |
| DAMO-YOLO-M [71] | 28.2 | 61.8 | 49.2 | 4.57 |
| DAMO-YOLO-L [71] | 42.1 | 97.3 | 50.8 | 6.48 |
| DEYO-M [45] | 33.0 | 78.0 | 50.7 | 7.14 |
| **YOLOv10-M** | **15.4** | **59.1** | **51.1** | **4.74** |
| YOLOv7 [62] | 36.9 | 104.7 | 51.2 | 17.03 |
| **YOLOv10-B** | **19.1** | **92.0** | **52.5** | **5.74** |
| YOLOv7-X [62] | 71.3 | 189.9 | 52.9 | 21.45 |
| DEYO-L [45] | 51.0 | 155.0 | 52.7 | 10.00 |
| **YOLOv10-L** | **24.4** | **120.3** | **53.2** | **7.28** |
| DEYO-X [45] | 78.0 | 242.0 | 53.7 | 15.38 |
| **YOLOv10-X** | **29.5** | **160.4** | **54.4** | **10.70** |

with our holistic efficiency-accuracy driven model design, showing a 2.0% AP improvement with only 0.05ms latency overhead. Besides, for YOLOv10-M (#6 in Tab. 2), which has a larger model scale and more redundancy, our efficiency driven model design results in a considerable 12.5% latency reduction, as shown in Tab. 2. When combined with accuracy driven model design, we observe a notable 0.8% AP improvement for YOLOv10-M, along with a favorable latency reduction of 0.48ms. These results well demonstrate the effectiveness of our design strategy across different model scales.

Table 19: Performance comparisons under 300 training epochs.

| Model | #Param.(M) | FLOPs(G) | $AP^{val}$(%) | Latency(ms) |
|---|---|---|---|---|
| YOLOv6-3.0-N [29] | 4.7 | 11.4 | 37.0 | 2.69 |
| Gold-YOLO-N [60] | 5.6 | 12.1 | 39.6 | 2.92 |
| **YOLOv10-N (Ours)** | **2.3** | **6.7** | 37.7 | **1.84** |
| YOLOv6-3.0-S [29] | 18.5 | 45.3 | 44.3 | 3.42 |
| Gold-YOLO-S [60] | 21.5 | 46.0 | 45.4 | 3.82 |
| YOLO-MS-XS [8] | 4.5 | 17.4 | 43.4 | 8.23 |
| **YOLOv10-S (Ours)** | **7.2** | **21.6** | 45.6 | **2.49** |
| YOLOv6-3.0-M [29] | 34.9 | 85.8 | 49.1 | 5.63 |
| Gold-YOLO-M [60] | 41.3 | 87.5 | 49.8 | 6.38 |
| **YOLOv10-M (Ours)** | **15.4** | **59.1** | 50.3 | **4.74** |
| YOLOv6-3.0-L [29] | 59.6 | 150.7 | 51.8 | 9.02 |
| Gold-YOLO-L [60] | 75.1 | 151.7 | 51.8 | 10.65 |
| YOLO-MS [8] | 22.2 | 80.2 | 51.0 | 12.41 |
| **YOLOv10-B (Ours)** | **19.1** | **92.0** | 51.6 | **5.74** |
| **YOLOv10-L (Ours)** | **24.4** | **120.3** | 52.4 | **7.28** |
| **YOLOv10-X (Ours)** | **29.5** | **160.4** | 53.6 | **10.70** |

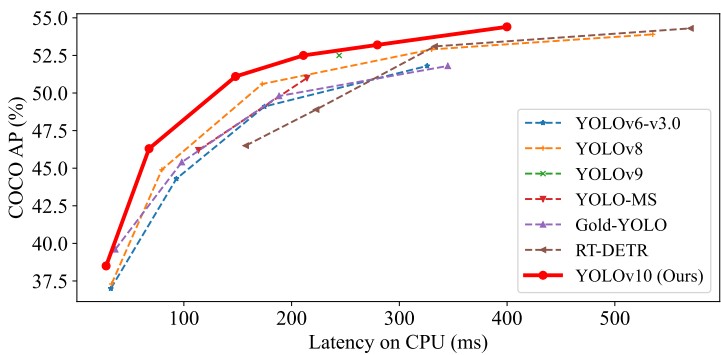

Figure 5: Performance and efficiency comparisons on CPU.

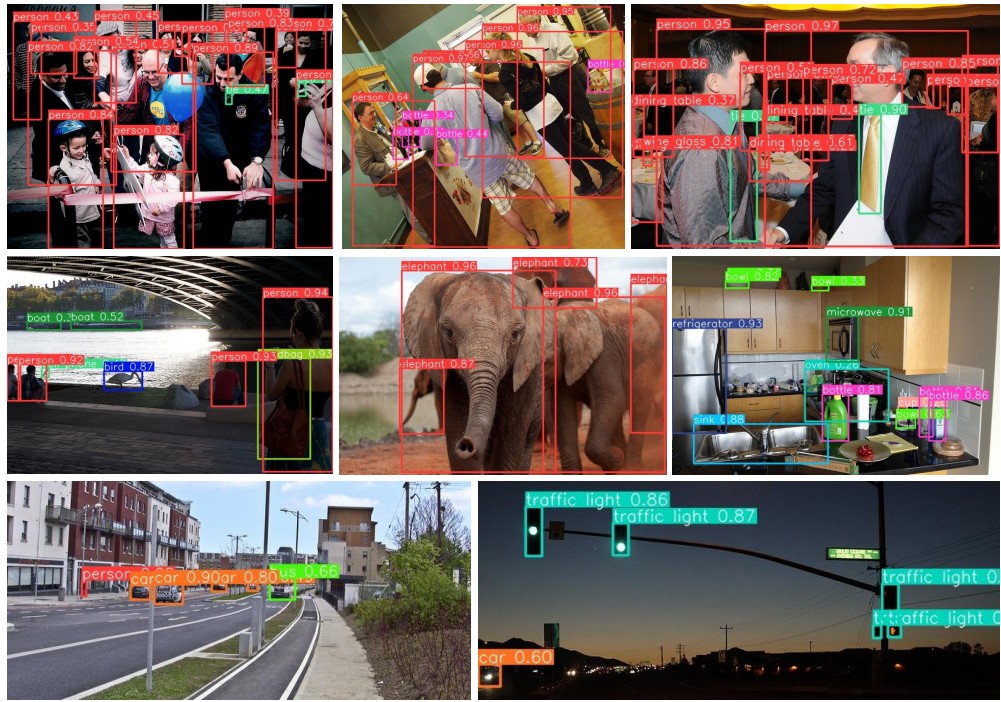

Figure 6: Visualization results under complex and challenging scenarios.

## A.8 Visualization Results

Fig. 6 presents the visualization results of our YOLOv10 in the complex and challenging scenarios. It can be observed that YOLOv10 can achieve precise detection under various difficult conditions, such as low light, rotation, *etc*. It also demonstrates a strong capability in detecting diverse and densely packed objects, such as bottle, cup, and person. These results indicate its superior performance.

## A.9 Contribution, Limitation, and Broader Impact

**Contribution.** In summary, our contributions are three folds as follows:

1. We present a novel consistent dual assignments strategy for NMS-free YOLOs. A dual label assignments way is designed to provide rich supervision by one-to-many branch during training and high efficiency by one-to-one branch during inference. Besides, to ensure the harmonious supervision between two branches, we innovatively propose the consistent matching metric, which can well reduce the theoretical supervision gap and lead to improved performance.

2. We propose a holistic efficiency-accuracy driven model design strategy for the model architecture of YOLOs. We present novel lightweight classification head, spatial-channel decoupled downsampling, and rank-guided block design, which greatly reduce the computational redundancy and achieve high efficiency. We further introduce the large-kernel convolution and innovative partial self-attention module, which effectively enhance the performance under low cost.

3. Based on the above approaches, we introduce YOLOv10, a new real-time end-to-end object detector. Extensive experiments demonstrate that our YOLOv10 achieves the state-of-the-art performance and efficiency trade-offs compared with other advanced detectors.

**Limitation.** Due to the limited computational resources, we do not investigate the pretraining of YOLOv10 on large-scale datasets, *e.g.*, Objects365 [52]. Besides, although we can achieve competitive end-to-end performance using the one-to-one head under NMS-free training, there still exists a performance gap compared with the original one-to-many training using NMS, especially noticeable in small models. For example, in YOLOv10-N and YOLOv10-S, the performance of one-to-many training with NMS outperforms that of NMS-free training by 1.0% AP and 0.5% AP, respectively. We will explore ways to further reduce the gap and achieve higher performance for YOLOv10 in the future work.

**Broader impact.** The YOLOs can be widely applied in various real-world applications, including medical image analyses and autonomous driving, *etc*. We hope that our YOLOv10 can assist in these fields and improve the efficiency. However, we acknowledge the potential for malicious use of our models. We will make every effort to prevent this.

