# OpenReview forum: "YOLOv10: Real-Time End-to-End Object Detection"
_NeurIPS.cc/2024/Conference — NeurIPS 2024 poster_

### Official Review · Reviewer_qWaL · 2024-06-13

**Soundness:** 3
**Presentation:** 2
**Contribution:** 3
**Rating:** 5
**Confidence:** 5

**Summary:**

This paper further advances the performance-efficiency boundary of YOLOs from both the post-processing and the model architecture. Specifically, the paper presents the consistent dual assignments for NMS-free training of YOLOs, which brings the competitive performance and low inference latency simultaneously. In addition, this paper performs engineering optimizations on components of YOLOs in terms of speed and accuracy to achieve optimal performance.

**Strengths:**

1. According to results in the paper, YOLOv10 achieves the best trade-off between speed and accuracy.
2. The idea of ​​removing NMS for YOLOs to improve is intuitive, and the advantages of NMS-free are analyzed in detail in the paper of RT-DETR[1].
3. The  the consistent dual assignments for NMS-free training is effective and significantly improves the end-to-end inference latency of YOLOs.
4. The holistic efficiency-accuracy driven optimizations listed in this paper is in line with the original design intention of the real-time detector.

[1] Zhao, Yian, et al. "Detrs beat yolos on real-time object detection." Proceedings of the IEEE/CVF Conference on Computer Vision and Pattern Recognition. 2024.

**Weaknesses:**

1. The motivation of this paper is not clear, and it seems to be intended to further optimize YOLOs from an engineering perspective to achieve better results. The analysis and solutions on NMS and model architecture are almost from existing technologies in the detection field. For example, Hbrid-DETR[1] proposed the drawbacks of one-to-one matching to achieve NMS-free, and then adopted a solution combining one-to-one matching and one-to-many matching; the results of this paper are inspiring, but more like a technical report, providing a stronger baseline for the field of real-time detection.
2. The paper claims that "the inherent complexity in deploying DETRs impedes its ability to attain the optimal balance between accuracy and speed." This seems to lack the necessary evidence. RT-DETR retains most of the components of DETRs and achieves strong results, greatly improving the detection speed and accuracy of DETRs and providing multiple deployment options.
3. If possible, please provide speed comparison results of YOLOv10 and competitors on other hardware.

[1] Jia, Ding, et al. "Detrs with hybrid matching." Proceedings of the IEEE/CVF conference on computer vision and pattern recognition. 2023.

**Questions:**

Please answer the questions stated in the weakness.

**Limitations:**

See Weaknesses.

---

> ### Author Rebuttal · Authors · 2024-08-05
>
> We sincerely appreciate your valuable feedback and insightful comments. Thank you for liking the speed-accuracy trade-offs of YOLOv10, the effectiveness of NMS-free training, and the design intention of efficiency-accuracy driven architectural optimizations. We provide the response to each comment in the point-to-point manner below. Please feel free to contact us if you have further concerns.
>
> **Q1-1: "The motivation of this paper is not clear. The analysis and solutions on NMS and model architecture are almost from existing technologies in the detection field." and " The results of this paper are inspiring, but more like a technical report."**
>
> **A1-1**: Thanks. We understand this concern. We think that there are some academic insights we wish to share with the community for further works on improving YOLOs. Therefore, we would like to discuss our motivation, observations, and contributions here. Our work seeks to improve YOLOs from the whole pipeline, including the post-processing and the architectural design. (1) To improve the post-processing step, similar to previous works [1], we find that the one-to-one matching, while eliminating the need for NMS, suffers from limited supervision and leads to inferior performance for YOLOs, as shown in Tab.3 in the paper. Therefore, we introduce dual label assignments including one-to-one matching and the original one-to-many matching for YOLOs to enrich the supervision during training and simplify the post-processing during inference. Furthermore, motivated by the observation that the supervision gap between two matching ways adversely affects the performance, as shown in Tab.4 and Fig.2.(b) in the paper, we propose the consistent matching metric for YOLOs to minimize the theoretical supervision gap. It brings improved supervision alignment and enhances performance without requiring hyper-parameter tuning. (2) To improve the architectural design, like previous works including ConvNeXt [2] and MobileLLM [3], we note that a systematical and comprehensive inspection for various components can further improve the efficiency and accuracy of YOLOs, which is lacking. This motivates us to holistically analyze and enhance the architectural designs. In addition to introducing existing designs, we also propose the new rank-guided block design and partial self-attention module to enhance the YOLOs. These efforts lead to favorable performance and efficiency improvements. We hope that the insights of our work can benefit the community and inspire further research. For example, we think that by simplifying the post-processing, the NMS-free training strategy leaves more room for the model itself, which may facilitate more advanced architectural optimizations and designs.
>
> **Q1-2: "For example, Hybrid-DETR[1] proposed the drawbacks of one-to-one matching and combined one-to-one matching and one-to-many matching."**
>
> **A1-2**: Thanks. While both having one-to-one and one-to-many matching, we think that the specific label assignment ways differ between Hybrid-DETR [1] and ours. Specifically, to achieve one-to-many matching, Hybrid-DETR introduces extra queries and repeats ground truth multiple times for bipartite matching, whereas we adopt the prediction-aware matching with the spatial prior of anchor points. Furthermore, differently, we theoretically analyze the supervision gap between two matching ways for YOLOs and present the consistent matching metric to reduce the supervision gap. It results in better performance without hyper-parameter tuning by improving the supervision alignment. We will incorporate more discussion with the mentioned work in the revision.
>
> **Q2: "The claim of "the inherent complexity in deploying DETRs impedes its ability to attain the optimal balance between accuracy and speed"."**
>
> **A2**: Thanks. We understand your concern regarding the clarity of the statement. We think that RT-DETR effectively propels DETRs into the realm of real-time detection. On the other hand, when only considering the forward process of model, we observe that its DETR-based architecture still lags behind the YOLO series models in terms of inference efficiency for certain model scales. For example, as shown in Tab. 1 in the paper, RT-DETR-R34 and RT-DETR-R18 show the higher latency of 1.12ms and 2.16ms compared with YOLOv8-M and YOLOv8-S respectively. We will revise the statement as below for more clear presentation in the revision: "when considering only the forward process of model during deployment, the efficiency of the DETRs still has room for improvements compared with YOLOs." If you have further concern regarding the revised statement, please let us know.
>
> **Q3: "Speed comparison on other hardware."**
>
> **A3**: Thanks. We present the speed comparison results of YOLOv10 and others on CPU (Intel Xeon Skylake, IBRS) using OpenVINO as below. We observe that YOLOv10 also shows superior performance and efficiency trade-offs. We will incorporate more results on other hardware if available, e.g., NPU, in future works.
>
> | Model| AP$^{val}$ (%)|Latency (ms)|
> |-------------|--------------|------------|
> |YOLOv6-3.0-N|37.0|32.22|
> |Gold-YOLO-N|39.6|36.35|
> |YOLOv8-N|37.3|32.91|
> |**YOLOv10-N**|38.5|27.84|
> |YOLOv6-3.0-S|44.3|93.11|
> |Gold-YOLO-S|45.4|98.08|
> |YOLO-MS-S|46.2|113.54|
> |YOLOv8-S|44.9|79.56|
> |RT-DETR-R18|46.5|157.40|
> |**YOLOv10-S**|46.3|67.70|
> |YOLOv6-3.0-M|49.1|174.82|
> |Gold-YOLO-M|49.8|188.33|
> |YOLO-MS|51.0|214.28|
> |YOLOv8-M|50.6|172.76|
> |RT-DETR-R34|48.9|222.82|
> |**YOLOv10-M**|51.1|147.99|
> |YOLOv6-3.0-L|51.8|325.85|
> |Gold-YOLO-L|51.8|344.99|
> |YOLOv9-C|52.5|244.14|
> |**YOLOv10-B**|52.5|210.98|
> |YOLOv8-L|52.9|330.02|
> |RT-DETR-R50|53.1|332.76|
> |**YOLOv10-L**|53.2|279.59|
> |YOLOv8-X|53.9|535.23|
> |RT-DETR-R101|54.3|570.25|
> |**YOLOv10-X**|54.4|399.87|
>
> [1] DETRs with Hybrid Matching, CVPR 2023.
>
> [2] A ConvNet for the 2020s, CVPR 2022.
>
> [3] MobileLLM: Optimizing Sub-billion Parameter Language Models for On-Device Use Cases,  ICML 2024.

---

### Official Review · Reviewer_snfm · 2024-06-25

**Soundness:** 3
**Presentation:** 3
**Contribution:** 3
**Rating:** 7
**Confidence:** 5

**Summary:**

This paper presents a one-stage object detector that introduces a consistent dual assignments strategy, effectively eliminating the need for NMS and significantly accelerating inference speed with minimal impact on accuracy. Additionally, the paper explores a series of model designs aimed at balancing efficiency and accuracy, enhancing overall model performance.

**Strengths:**

1. The introduction of a consistent dual assignments strategy enables the one-stage detector to operate without NMS, significantly boosting inference speed.
2. The focus on efficiency-driven model design minimizes accuracy loss while improving the model's efficiency.
3. The incorporation of large-kernel convolutions and partial self-attention mechanisms enhances model performance with minimal additional computational cost.

**Weaknesses:**

1. The paper could benefit from discussing related works on hybrid or multiple group label assignment strategies, which are prevalent in recent DETR-based detectors.

2. The implementation of a one-to-many head as an auxiliary head likely increases training costs. The paper lacks comparative data on training time and costs, which should be addressed to provide a comprehensive evaluation of the model's efficiency.

**Questions:**

See weakness. Besides, you mention that intrinsic ranks indicate redundancy, leading to the replacement of certain blocks with CIB in high redundancy stages. After these modifications, do the intrinsic ranks in these stages actually decrease?

**Limitations:**

The paper discusses their limitations in the appendix.

---

> ### Author Rebuttal · Authors · 2024-08-05
>
> We sincerely appreciate your valuable feedback and insightful comments. Thank you for liking the consistent dual assignments strategy, efficiency-accuracy driven model designs, and the improved performance and speed. We provide the response to each comment in the point-to-point manner below. Please feel free to contact us if you have further concerns.
>
> **Q1: "The paper could benefit from discussing related works on hybrid or multiple group label assignment strategies, which are prevalent in recent DETR-based detectors."**
>
> **A1**: Thanks for this suggestion. We will incorporate more discussion with the related works in the revision. In improving upon DETR, previous works, including H-DETR [1], Group-DETR [2], MS-DETR [3], etc., introduce one-to-many matching in conjunction with the original one-to-one matching by hybrid or multiple group label assignments, to accelerate the training convergence and improve the performance. For example, H-DETR introduces the hybrid branch/epoch/layer schemes, Group-DETR presents group-wise one-to-many assignment, and MS-DETR imposes one-to-many supervision on queries of the primary decoder. Similarly, we introduce dual label assignments including the one-to-one matching and the original one-to-many matching for YOLOs, to provide rich supervision during training and high efficiency during inference. Differently, to achieve the one-to-many matching, they usually introduce extra queries or repeat ground truths for bipartite matching, or select top several queries from the matching scores, while we adopt the prediction aware assignment that incorporates the spatial prior. Besides, we analyze the supervision gap between the two heads and present the consistent matching metric for YOLOs to reduce the theoretical supervision gap. It improves performance through better supervision alignment and eliminates the need for hyper-parameter tuning.
>
> **Q2: "The implementation of a one-to-many head as an auxiliary head likely increases training costs. The paper lacks comparative data on training time and costs, which should be addressed to provide a comprehensive evaluation of the model's efficiency."**
>
> **A2**: Thanks. We measure the training cost on 8 NVIDIA 3090 GPUs and present the results with and without the auxiliary head based on YOLOv10-M as below. We also compare our training cost with other detectors using the official codebases. We observe that benefiting from the lightweight design, introducing the auxiliary head results in a small increase in the training time, about 18s each epoch, which is affordable. To fairly verify the benefit of the auxiliary head, we also adopt longer 550 training epochs for YOLOv10-M w/o auxiliary head, which leads to a similar training time (29.3 vs. 29.1 hours) but still yields inferior performance (48.9% vs. 51.1% AP) compared with the model w/ auxiliary head. Besides, we also note that YOLOv10 shows a relatively low training cost compared with others. We will incorporate more results and analyses about the training time and costs in the revision.
>
> |Model|Training Time (min/epoch)|Epochs|Total Time (hour)|
> |--------------------------------|-------------------------|------|-----------------|
> |YOLOv6-3.0-M|8.3|300|41.7|
> |YOLOv8-M|3.3|500|27.3|
> |YOLOv9-M|4.9|500|40.7|
> |Gold-YOLO-M|12.8|300|63.8|
> |YOLO-MS|8.5|300|42.3|
> |**YOLOv10-M** w/o auxiliary head|3.2|500|26.7|
> |**YOLOv10-M** w/ auxiliary head|3.5|500|29.1|
>
> **Q3: "Besides, you mention that intrinsic ranks indicate redundancy, leading to the replacement of certain blocks with CIB in high redundancy stages. After these modifications, do the intrinsic ranks in these stages actually decrease?"**
>
> **A3**: Thanks. We would like to clarify that a lower intrinsic rank indicates higher redundancy, and a higher rank implies more condensed parameters. For example, given a parameter matrix $W$, a low intrinsic rank suggests that it can be replaced through low-rank approximation with fewer parameters. Such an approximation is not to improve the rank of the original $W$, but to reduce its redundancy of parameters. Similarly for our work, we explore the intrinsic rank to guide the replacement with CIB in the high redundancy stages, to reduce the computational costs while maintaining the model capacity. As shown in Tab. 5 in the paper, it can result in the competitive performance but with fewer parameters and less computation. In Fig.2 in the global response pdf, we visualize the intrinsic ranks after the modifications. For the CIB block, we calculate the numerical rank of the last 1\*1 convolution with major parameters. We observe that the resulting models with fewer parameters exhibit the similar ranks to before, thereby maintaining the competitive capacity compared with the original models, while enjoying faster inference speed. We also think that improving the intrinsic rank to obtain better effectiveness and efficiency is an interesting idea and we leave it for the future work.
>
> [1] DETRs with Hybrid Matching, CVPR 2023
>
> [2] Group DETR: Fast DETR Training with Group-Wise One-to-Many Assignment, ICCV 2023.
>
> [3] MS-DETR: Efficient DETR Training with Mixed Supervision, CVPR 2024.

---

> > ### Comment · Reviewer_snfm · 2024-08-09
> >
> > Thank you for addressing my concerns. With the clarifications provided, I am satisfied and will maintain my current evaluation score.

---

### Official Review · Reviewer_nrqj · 2024-07-11

**Soundness:** 3
**Presentation:** 3
**Contribution:** 2
**Rating:** 5
**Confidence:** 4

**Summary:**

The authors further improve the YOLO series detector to the tenth generation, through the following two points. The first part is the post-processing improvement, and the authors propose consistent dual assignments for NMS-free training. The second part is the effective model architecture design, where the authors reconstruct the internal structure details. Comprehensive experiments are performed to validate that YOLOv10 strikes a balance between efficiency and accuracy well.

**Strengths:**

1. The authors propose a good lightweight detection framework and effectively push the YOLO framework to the End-to-End inference paradigm, which is undoubtedly admirable.
2. The manuscript is explicit and well-organized.
3. Experimental validation is sufficient and solid. The authors conduct comprehensive experiments on various metrics and show a large improvement, which is impressive. Furthermore, the ablation study is in detail.

**Weaknesses:**

1. The novelty of the paper is weak. First, the “dual assignments” is similar to H-DETR [1], where the hybrid matching scheme has been proposed, and is consistent with the dual assignment of LRANet [2]. Besides, the lightweight classification head and spatial-channel decoupled downsampling have been proposed by DAMO-YOLO [3] and large-kernel convolution is common in previous methods. The paper is more like a technical report.
2. More lightweight detectors should be listed in Table 1, including DAMO-YOLO, and YOLOv7. Besides, the authors should specify at what batch size the delay is calculated.

[1] Jia, Ding, et al. "Detrs with hybrid matching." *Proceedings of the IEEE/CVF conference on computer vision and pattern recognition*. 2023.

[2] Su, Yuchen, et al. "LRANet: Towards Accurate and Efficient Scene Text Detection with Low-Rank Approximation Network." *Proceedings of the AAAI Conference on Artificial Intelligence*. Vol. 38. No. 5. 2024.

[3] Xu, Xianzhe, et al. "Damo-yolo: A report on real-time object detection design." *arXiv preprint arXiv:2211.15444* (2022).

**Questions:**

1. This End-to-End paradigm is still based on the anchor-based dense detection paradigm, so the final prediction results have a lot of background. Usually, there are only less than 300 objects in a picture, so there will be almost 8000 invalid detections in YOLOs, which is undoubtedly a waste. In terms of efficiency, I think it is a bit inferior to the object query of the DETR framework.

**Limitations:**

The authors adequately addressed the limitations.

---

> ### Author Rebuttal · Authors · 2024-08-05
>
> We sincerely appreciate your valuable feedback and insightful comments. Thank you for liking the end-to-end framework, well-organized writing, and solid experimental validation. We provide the response to each comment in the point-to-point manner below. Please feel free to contact us if you have further concerns.
>
> **Q1-1: "The “dual assignments” is similar to H-DETR and LRANet. "**
>
> **A1-1**: Thanks. While there are structural similarities in the dual assignments among H-DETR, LRANet, and ours, we think that the specific label assignment mechanisms employed differ between ours and the mentioned works. Specifically, LRANet employs the dense assignment and sparse assignment branches, which all belong to the one-to-many assignment, while we adopt the one-to-many and one-to-one branches. H-DETR introduces extra queries and repeats the ground truth several times to perform one-to-many matching using bipartite matching, while we leverage the prediction aware one-to-many matching with the spatial prior of anchor points. Furthermore, differently, we introduce the consistent matching metric for YOLOs, aiming to narrow the theoretical supervision gap between the one-to-many and one-to-one branches. It effectively improves the supervision alignment between two branches and results in performance enhancement without hyper-parameter tuning. We will incorporate more discussions with the mentioned works in the revision.
>
> **Q1-2: "The lightweight classification head and spatial-channel decoupled downsampling have been proposed by DAMO-YOLO"**
>
> **A1-2**: Thanks. While our lightweight classification head and spatial-channel decoupled downsampling have structural similarities to the ZeroHead and MobBlock used in DAMO-YOLO, we think that they are motivated by different considerations and have distinct architectural designs. Specifically, the ZeroHead in DAMO-YOLO is intended to balance the computational cost between the neck and head by using a single linear layer for both classification and regression tasks. In contrast, our approach addresses the redundancy in the classification head relative to the regression head by proposing a lightweight design for the classification task. Moreover, DAMO-YOLO applies the MobBlock only to the lightweight model through NAS, which also results in the identical structure for the basic block and downsampling module. Differently, we simplify the downsampling process for all model scales to reduce the computational redundancy, which is also decoupled from the basic block design. We understand this concern because of the similarity. However, we also introduce other new architectural designs for YOLOs including rank-guided block design and partial self-attention module, which help to achieve better performance and efficiency.
>
> **Q1-3: "Large-kernel convolution is common in previous methods."**
>
> **A1-3**: Thanks. Large kernel convolution is widely adopted in previous works due to its ability to effectively increase the receptive field. In contrast to prior studies, our findings indicate that the benefits of large kernel convolution vary across different model scales of YOLOs. Specifically, we observe that it provides performance gains for small-scale models but no improvements for larger models. Consequently, we use large kernel convolutions based on the model size, achieving a better efficient-accuracy trade-off. Besides, we also introduce the new architectural designs for YOLOs, including the rank-guided block design and partial self-attention module, which bring favorable benefits.
>
> **Q1-4: "The paper is more like a technical report."**
>
> **A1-4**: Thanks. We understand this concern. We think that there are some academic inspirations we want to share with the community, especially for YOLO research. Therefore, we would like to revisit our motivation, findings, and contributions here. Please refer to the **Discussion 1** in the global response for the specific content.
>
> **Q2-1: "More lightweight detectors should be listed in Table 1, including DAMO-YOLO, and YOLOv7."**
>
> **A2-1**: Thanks. We present the comparisons with more lightweight detectors, including DAMO-YOLO and YOLOv7, as below. We will incorporate these results in the revision.
>
> |Model|Param. (M)|FLOPs (G)|AP$^{val}$ (%)|Latency (ms)|
> |-------------|----------|---------|--------------|------------|
> |DAMO-YOLO-T|8.5|18.1|42.0|2.21|
> |DAMO-YOLO-S|16.3|37.8|46.0|3.18|
> |**YOLOv10-S**|7.2|21.6|46.3|2.49|
> |DAMO-YOLO-M|28.2|61.8|49.2|4.57|
> |DAMO-YOLO-L|42.1|97.3|50.8|6.48|
> |**YOLOv10-M**|15.4|59.1|51.1|4.74|
> |YOLOv7|36.9|104.7|51.2|17.03|
> |**YOLOv10-B**|19.1|92.0|52.5|5.74|
> |YOLOv7-X|71.3|189.9|52.9|21.45|
> |**YOLOv10-L**|24.4|120.3|53.2|7.28|
>
> **Q2-2: "What batch size the delay is calculated"**
>
> **A2-2**: Thanks. We follow previous works [1,2] and adopt the batch size of 1 for measuring the delay.
>
> **Q3: "Discussion with the DETR framework."**
>
> **A3**: Thanks for your insightful discussion. Considering that the YOLOs and the DETR framework are two important topics as independent techniques for modern object detection, a comprehensive comparison between them is challenging yet highly interesting and meaningful. We think that this is a trade-off in the design of detectors and they may complement each other and advance together. On the one hand, DETR enjoys fewer invalid predictions compared with the anchor-based detection paradigm by utilizing the object query design, which shows less waste. On the other hand, the anchor-based detection paradigm directly outputs predictions without introducing the transformer decoder to interact with object queries, which simplifies the deployment. Given that both paradigms have the respective following works in the community, we believe that the best of these two paradigms may complement each other to achieve higher efficiency and performance in future works.
>
> [1] DETRs Beat YOLOs on Real-time Object Detection, CVPR 2024.
>
> [2] YOLOv8.

---

> ### Comment · Reviewer_nrqj · 2024-08-08
>
> Thank you for your kind and detailed responses and most of my concerns are now resolved. Although I still maintain a cautious attitude toward the innovative aspect of this paper, I would like to keep my positive score for the submission.

---

### Official Review · Reviewer_vpxY · 2024-07-11

**Soundness:** 3
**Presentation:** 3
**Contribution:** 3
**Rating:** 6
**Confidence:** 4

**Summary:**

The paper presents YOLOv10, an advancement in the YOLO series for real-time object detection. The authors have focused on eliminating the need for Non-Maximum Suppression (NMS) and optimizing the model architecture for efficiency and accuracy. The paper includes a novel training strategy and architectural enhancements, demonstrating improved performance on the COCO dataset.

**Strengths:**

1. The introduction of consistent dual assignments for NMS-free training is an innovative solution that potentially improves the end-to-end deployment efficiency of object detectors.
2. The paper presents a comprehensive strategy for optimizing model components, which is a significant advancement over prior works that focused on isolated aspects of model design.
3. The paper is backed by extensive empirical evidence, showing improvements in both speed and accuracy over previous models.

**Weaknesses:**

1.  While the paper introduces a new training strategy, the innovations in the model's architectural design are not as pronounced. A more detailed exposition of the creative process and rationale behind the architectural choices would strengthen the paper's contribution to the field.
2. The paper acknowledges a 1% Average Precision (AP) difference when comparing the YOLOv10-N model trained with NMS-free to the original one-to-many training with NMS. It is better to provide further performance analysis of NMS-free training.
3. Latency of YOLOv10 with the original one-to-many training using NMS should be reported.

**Questions:**

1. Could you report more details on the training cost compared to other YOLOs, such as total training epochs and throughput( with dual label assignments training).
2.  Given that previous iterations of the YOLO series have typically been trained for 300 epochs, would the authors be able to present the training outcomes for YOLOv10 at this epoch count?

**Limitations:**

This paper has discussed the limitations in Appendix.

---

> ### Author Rebuttal · Authors · 2024-08-05
>
> We sincerely appreciate your valuable feedback and insightful comments. Thank you for liking the new training strategy, comprehensive model optimization strategy, and extensive experiments. We provide the response to each comment in the point-to-point manner below. Please feel free to contact us if you have further concerns.
>
> **Q1: "While the paper introduces a new training strategy, the innovations in the model's architectural design are not as pronounced. A more detailed exposition of the creative process and rationale behind the architectural choices would strengthen the paper's contribution to the field."**
>
> **A1**: Thank you for liking our NMS-free training strategy. Like ConvNeXt [1] and MobileLLM [2], our architectural choices are motivated by a comprehensive inspection for various components, systematically considering both efficiency and accuracy aspects. While the architectural designs may appear less pronounced compared with the proposed training strategy, they are not equipped without any rationale. For example, our lightweight classification head design is driven by reducing the computational redundancy in the classification task relative to the regression task. Our rank-guided block design is motivated by adopting the compact block structure adaptively based on the redundancy indicated by intrinsic ranks. We will make them more detailed and comprehensive in the revision. Besides, we believe that benefiting from the NMS-free training strategy, there is greater flexibility in the architectural designs, which can inspire further research on the YOLO architecture. Therefore, our model design strategies can serve as an initial exploration that paves the way for more advanced designs in the future.
>
> **Q2: "It is better to provide further performance analysis of NMS-free training."**
>
> **A2**: Thanks. In Tab.1 in the paper, we observe that the performance gap between NMS-free training and the original one-to-many training diminishes as the model size increases. Therefore, we can reasonably conclude that such a gap can be attributed to the limitations in the model capability. Notably, unlike the original one-to-many training using NMS, the NMS-free training necessitates more discriminative features for one-to-one matching. In the case of the YOLOv10-N model, its limited capacity results in extracted features that lack sufficient discriminability, leading to a more notable performance gap. In contrast, the YOLOv10-X model, which possesses stronger capability and more discriminative features, shows no performance gap between two training strategies. In Fig.1 in the global response pdf, we visualize the average cosine similarity of each anchor point's extracted features with those of all other anchor points on the COCO validation set. We observe that as the model size increases, the feature similarity between anchor points exhibits a downward trend, which benefits the one-to-one matching. We will incorporate these results in the revision.
>
> **Q3: "Latency of YOLOv10 with the original one-to-many training using NMS should be reported."**
>
> **A3**: Thanks. We measure the latency of YOLOv10 with the original one-to-many training using NMS and report the results below.
>
> |Model|Latency|
> |---------|-------|
> |YOLOv10-N|6.19ms|
> |YOLOv10-S|7.15ms|
> |YOLOv10-M|9.03ms|
> |YOLOv10-B|10.04ms|
> |YOLOv10-L|11.52ms|
> |YOLOv10-X|14.67ms|
>
> **Q4: "More details on the training cost compared to other YOLOs, such as total training epochs and throughput (with dual label assignments training)."**
>
> **A4**: Thanks.  We measure the training throughput on 8 NVIDIA 3090 GPUs using the official codebases and present the comparison results based on the medium variant below. We observe that despite having 500 training epochs, YOLOv10 achieves a high training throughput, making its training cost affordable.
>
> |Model|Epochs|Throughput (epochs/hour)|Total Time (hour)|
> |-------------|------|------------------------|-----------------|
> |YOLOv6-3.0-M|300|7.2|41.7|
> |YOLOv8-M|500|18.3|27.3|
> |YOLOv9-M|500|12.3|40.7|
> |Gold-YOLO-M|300|4.7|63.8|
> |YOLO-MS|300|7.1|42.3|
> |**YOLOv10-M**|500|17.2|29.1|
>
> **Q5: "Training outcomes for YOLOv10 at 300 epochs."**
>
> **A5**: Thanks. In experiments, we follow previous works [3,4] to train the models for 500 epochs. Here, we conduct experiments to train the models for 300 epochs and present the comparison results with YOLOv6[5], Gold-YOLO[6], and YOLO-MS[7] which adopt 300 epochs as below. We observe that YOLOv10 also achieves superior performance and efficiency. Besides, we note that despite trained for 500 epochs, YOLOv10 has less training cost compared with these models as presented in **A4**.
>
> |Model|AP$^{val}$ (%)|Latency (ms)|
> |-------------|--------------|------------|
> |YOLOv6-3.0-N|37.0|2.69|
> |Gold-YOLO-N|39.6|2.92|
> |**YOLOv10-N**|37.7|1.84|
> |YOLOv6-3.0-S|44.3|3.42|
> |Gold-YOLO-S|45.4|3.82|
> |YOLO-MS-XS|43.4|8.23|
> |**YOLOv10-S**|45.6|2.49|
> |YOLOv6-3.0-M|49.1|5.63|
> |Gold-YOLO-M|49.8|6.38|
> |**YOLOv10-M**|50.3|4.74|
> |YOLOv6-3.0-L|51.8|9.02|
> |Gold-YOLO-L|51.8|10.65|
> |YOLO-MS|51.0|12.41|
> |**YOLOv10-B**|51.6|5.74|
> |**YOLOv10-L**|52.4|7.28|
> |**YOLOv10-X**|53.6|10.70|
>
> [1] A ConvNet for the 2020s, CVPR 2022.
>
> [2] MobileLLM: Optimizing Sub-billion Parameter Language Models for On-Device Use Cases,  ICML 2024.
>
> [3] YOLOv8.
>
> [4] YOLOv9: Learning What You Want to Learn Using Programmable Gradient Information, arXiv preprint 2024.
>
> [5] YOLOv6 v3.0: A Full-Scale Reloading, arXiv preprint 2023.
>
> [6] Gold-YOLO: Efficient object detector via gather-and-distribute mechanism, NeurIPS 2023.
>
> [7] YOLO-MS: YOLO-MS: rethinking multi-scale representation learning for real-time object detection, arXiv preprint 2023.

---

> > ### Comment · Reviewer_vpxY · 2024-08-08
> >
> > Thank you for your response, it solved my concerns and I will change my rating to 6.

---

### Author Rebuttal · Authors · 2024-08-05

**Discussion 1: About the technical novelty.**

Thanks. We would like to discuss more about our motivation, findings, and contributions.

Our motivation stems from optimizing the detection pipeline of YOLOs, including the NMS post-processing and model architectural design. (1) For the post-processing, we find that simply adopting the one-to-one matching for YOLOs results in notable performance degradation due to the limited supervision signals, as shown in Tab.3 in the paper. This motivates us to introduce dual label assignments for YOLOs to enrich the supervision during training and eliminate the need for NMS during inference simultaneously. We further observe that the supervision gap between the two branches leads to inferior performance, as shown in Fig. 2.(b) and Tab.4 in the paper. To address this, we theoretically analyze the supervision gap and present the consistent matching metric for YOLOs to improve the supervision alignment. It leads to favorable performance enhancement. (2) For the model architectural design, we note that a comprehensive inspection of various components in YOLOs from both efficiency and accuracy aspects is lacking. This motivates us to follow previous works like ConvNeXt [1] and MobileLLM [2], to systematically analyze and improve architectural designs. In addition to introducing the advanced design strategies, we also present the new rank-guided block design and partial self-attention module for YOLOs. The resulting model architecture exhibits favorable efficiency and accuracy gains.

Besides, we think that the NMS-free training strategy leaves more room for further research in optimizing YOLO models from different aspects, including the architecture designs. The introduced architectural optimization can be regarded as an initial exploration toward efficient YOLO architecture. Therefore, we believe that our work can serve as a strong baseline and inspire further advancements.

[1] A ConvNet for the 2020s, CVPR 2022.

[2] MobileLLM: Optimizing Sub-billion Parameter Language Models for On-Device Use Cases,  ICML 2024.

---

### Decision · Program_Chairs · 2024-09-25

**Decision:**

Accept (poster)

**Comment:**

This paper proposes YOLOv10, an advancement of the real-time YOLO series object detection methods. The reviewers have a consensus that the proposed method is effective, and the experiments are sufficient and solid, demonstrating impressive performance improvements. All reviewers acknowledged that their concerns were addressed by the authors in the rebuttal phase. The AC agrees with the reviewers and recommends Accept for this paper.